# Spatial characterization of near-surface structure and meltwater runoff conditions across Devon Ice Cap from dual-frequency radar reflectivity

Kristian Chan[1], Cyril Grima[1], Anja Rutishauser[2], Duncan A. Young[1], Riley Culberg[3], Donald D. Blankenship[1]

[1]Institute for Geophysics, University of Texas at Austin, Austin, TX, 78758, USA
[2]Geological Survey of Denmark and Greenland, Copenhagen, Denmark
[3]Department of Electrical Engineering, Stanford University, Stanford, CA, 94305, USA

*Correspondence to*: Kristian Chan (kristian.chan@utexas.edu)

**Abstract.** Melting and refreezing processes in the firn of Devon Ice Cap control meltwater infiltration and runoff across the ice cap, but their full spatial extent and effect on near-surface structure is difficult to measure with surface-based traverses or existing satellite remote sensing. Here, we derive the coherent component of the near-surface return from airborne ice-penetrating radar surveys over Devon Ice Cap, Canadian Arctic, to characterize firn containing centimeter to meter-thick ice layers (i.e., ice slabs) formed from refrozen meltwater in firn. We assess the use of dual-frequency airborne ice-penetrating radar to characterize the spatial and vertical near-surface structure of Devon Ice Cap, by leveraging differences in range resolution of the radar systems. Comparison with reflectivities using a thin layer reflectivity model, informed by surface-based radar and firn core measurements, indicate that the coherent component is sensitive to the near-surface firn structure composed of quasi-specular ice and firn layers, limited by the bandwidth-constrained radar range resolution. Our results suggest that average ice slab thickness throughout the Devon Ice Cap percolation zone ranges from 4.2 to 5.6 m. This implies conditions that can enable lateral meltwater runoff and potentially contribute to the total surface runoff routed through supraglacial rivers down glacier. Together with the incoherent component of the surface return previously studied, our dual-frequency approach provides an alternative method for characterizing bulk firn properties, particularly where high resolution radar data are not available.

## 1 Introduction

Over the past several decades, the threat of sea level rise has led to increased attention on how Earth's polar regions respond to climate change. Warmer temperatures have led to increasing surface meltwater on glaciers and ice sheets in the Arctic (Mortimer et al., 2016; Trusel et al., 2018). However, the contribution of surface meltwater runoff to surface mass balance is complicated by the potential of meltwater refreezing in firn with sufficient cold content (i.e., the energy required to bring firn to its melting temperature), thereby buffering mass loss (van den Broeke et al., 2016; MacFerrin et al., 2019; Vandecrux et al., 2020a). The processes governing melting and refreezing are complex and remain challenging to capture in firn models,

which yield large discrepancies between estimates of meltwater retention (Vandecrux et al., 2020b). For example, refreezing releases latent heat, which subsequently induces local changes in temperature, thermal conductivity, and porosity in the firn column (Bezeau et al., 2013; van den Broeke et al., 2016). Meltwater that refreezes within the firn pore space typically form ice layers ranging from <0.1 m thick 'ice lenses' to meters-thick 'ice slabs' perched just below the surface (MacFerrin et al.,

2019). While firn remains relatively permeable in the presence of discontinuous thin ice lenses, the aggregation of ice lenses into horizontally continuous, low-permeability ice slabs over multiple seasons could in effect limit vertical meltwater percolation, thus promoting lateral runoff that could further contribute to sea level rise (Machguth et al., 2016; MacFerrin et al., 2019). Therefore, characterizing the effects of surface meltwater infiltration and refreezing on the firn's storage capacity is crucial for predicting the future runoff budget in response to increased climate warming.


Devon Ice Cap (DIC), located in the Canadian Arctic, has experienced intense seasonal surface melting since the mid-2000s, resulting in increased meltwater percolation in firn and the formation of ice slabs over time (Gascon et al., 2013a, b). Thus, DIC is well suited for studying the mechanisms that control firn hydrology, with potential applications to other regions also experiencing significant surface melting such as Greenland. Constraining the thickness and spatial extent of ice slabs would

provide insight into the potential for enhanced lateral meltwater runoff throughout DIC.

Field-based techniques have expanded our knowledge of melting and refreezing processes in firn (Bell et al., 2008; Sylvestre et al., 2013; Forster et al., 2014; Gascon et al., 2014; Machguth et al., 2016), and their observations are also necessary to help validate firn models that are often relied upon for mass balance estimates (Ashmore et al., 2020). Field studies typically

utilize some combination of firn cores, snow pits, and surface-based radar measurements, which provide in-situ measurements or high resolution remotely sensed observations of the local firn column. However, such methods often lack the capability to study spatially extensive regions or ones difficult to access. Thus, surveys with large spatial coverage are especially important for characterizing the spatial extent of melting and refreezing processes in firn.

Airborne ice-penetrating radar (IPR) (also known as radio echo sounding) surveys conducted at ultra high frequency (UHF) and higher have successfully directly mapped melt layers and ice slabs in firn with extensive spatial coverage compared to surface-based methods (Arnold et al., 2019; MacFerrin et al., 2019; Culberg et al., 2021). However, most available airborne ice-penetrating radar data collected at VHF frequencies and have commensurately lower bandwidths that limit their ability to resolve layers in the firn that are thinner than the radar range resolution (typically about 5-10 m, see Sec. 2.1) (Arnold et al.,

2018). Despite these challenges, airborne IPRs operating at VHF frequencies have been successfully used to characterize near-surface properties (Rutishauser et al., 2016; Grima et al., 2014b, 2016, 2019) and can be of benefit in regions lacking higher frequency radar coverage. Similar studies conducted at even lower frequencies have been used to study near-surface properties on Mars, such as thin deposits that cannot be directly resolved (Mouginot et al., 2009; Grima et al., 2012). Thus,

methods applied to lower frequency radar data would be useful for interpreting data from future Earth observing and planetary missions equipped with radar sounders.

Here, we apply the Radar Statistical Reconnaissance (RSR) method (Grima et al., 2014a) to airborne IPR datasets collected over Devon Ice Cap including the Operation IceBridge (MacGregor et al., 2021) and the SEARCH[Arctic] (SRH1) project (Rutishauser et al., 2022), as well as the initial 2014 Greenland Outlet Glaciers (GOG3) survey (Rutishauser et al., 2018, 2016). RSR is a statistical method that links the distribution of surface echo amplitudes to an analytical probability density function in order to derive the coherent and incoherent components of the total surface power. Rutishauser et al. (2016) previously used the incoherent surface power derived from the sparse GOG3 survey to map firn zones over DIC. These zones are distinguished by the presence of thin ice lenses (Zone I), thick ice slabs (Zone II), and predominantly glacier ice (Zone III). In this study, we use the ratio of the coherent to incoherent power to redefine these zones, derived solely from the SRH1 dataset providing greater spatial coverage compared to the GOG3 survey (Fig. 1). We then evaluate the combined use of airborne IPRs operating at two different frequencies and bandwidths for characterizing the vertical and spatial near-surface structure, both locally by zone and holistically across the ice cap. We compare the coherent power to modeled reflectivity values using a thin layer reflectivity model (Born and Wolf, 1970) along a profile with collocated surface-based radar data, to validate our interpretation of the coherent power derived from the airborne IPR datasets. Although these airborne radar systems cannot resolve layers thinner than the radar range resolution, we show how they can still modulate the surface coherent power. Finally, hypotheses about the near-surface hydrological conditions across zones are discussed based on our interpretations of the airborne IPR datasets, supported by imagery on DIC.

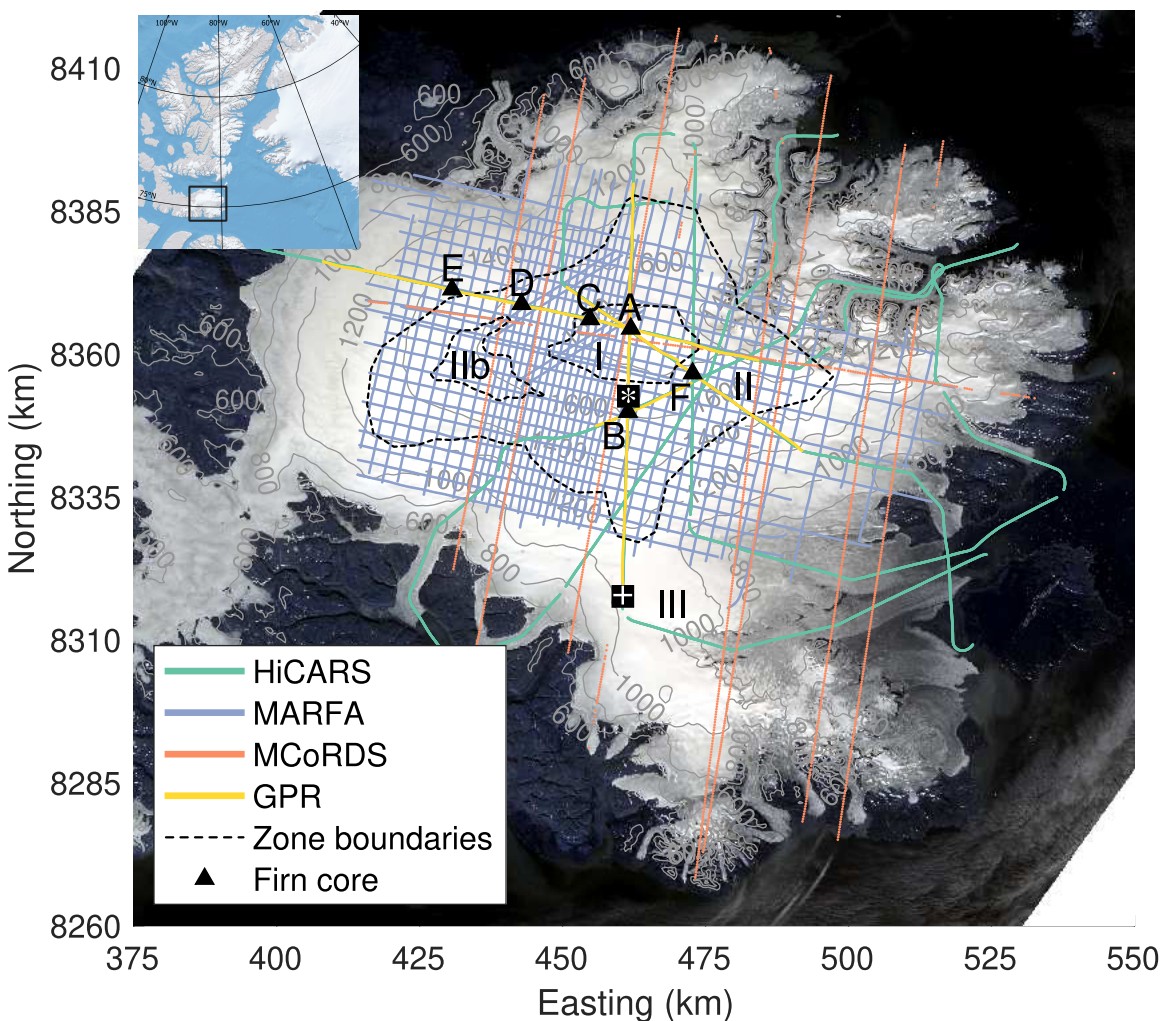

Figure 1. Map of airborne ice-penetrating radar data (HiCARS2 in 2014, MARFA in 2018, and MCoRDS3 in 2019) and surface-based radar data (in 2015) over Landsat image of Devon Ice Cap (courtesy of the U.S. Geological Survey). Location of the ice cap in the Arctic is depicted in the upper-left corner. Firn zones are marked with roman numerals. Thin gray lines indicate elevation contours displayed from 600 m to 2000 m every 200 m. The segment from '*' to '+' indicates the profile in Figure 4.

## 2 Data and Methods

### 2.1 Airborne ice-penetrating radar

Airborne IPR data were collected over DIC in 2014 with the High-Capability Airborne Radar Sounder 2 (HiCARS2) (Peters et al., 2007; Rutishauser et al., 2016, 2018) and in 2018 with the Multifrequency Airborne Radar-sounder for Full-phase Assessment (MARFA) (Scanlan et al., 2020; Rutishauser et al., 2022), which is a dual-phase center version of the HiCARS2 system (Fig. 1). Both instruments, operated by the University of Texas Institute for Geophysics, transmit a chirp with a center frequency ($f_c$) of 60 MHz and bandwidth of 15 MHz. In addition, airborne IPR data collected with the Multichannel

Coherent Radar Depth Sounder 3 (MCoRDS3), operated by the University of Kansas Center for Remote Sensing and Integrated Systems (CReSIS) for Operation IceBridge in 2019 was used for this study (Fig. 1), at a center frequency of 195 MHz and bandwidth of 30 MHz (MacGregor et al., 2021; Rodriguez-Morales et al., 2014).

The radar return from the surface is influenced to a depth that is equal to the vertical/range resolution, which we can also refer to as the near-surface depth $z_0$. The range resolution, and thus $z_0$, is calculated as (Cavitte et al., 2021)

$$z_0 = \frac{kc}{2B\sqrt{\varepsilon_{eff}}},$$
(1)

where $B$ is the radar bandwidth, $c$ is the speed of light in vacuum, $k$ is the windowing factor, and $\varepsilon_{eff}$ is the relative effective permittivity of the target medium (Table 1). For MCoRDS3, $k = 1.53$ as a result of the 20% Tukey time-domain window and Hanning frequency-domain window applied when performing pulse compression (CReSIS, 2016). For HiCARS2/MARFA, we adopt a factor of $k = 1.515$, as the ratio of the 100 ns compressed pulse width used in practice (Cavitte et al., 2021) to the theoretical 66 ns obtained from the 15 MHz bandwidth. $\varepsilon_{eff}$ of the near-surface is derived from

the relative permittivity ($\varepsilon$) of the individual components of a multi-component heterogeneous medium. Various material properties influence the permittivity, such as inclusion geometry, density, impurities, temperature, and ice crystal fabric orientation (Sihvola, 1999; Fujita et al., 2000; Pettinelli et al., 2015). In the case of a layered medium, the near-surface can be represented with prescribed layer thicknesses and permittivities (see Sec 2.3). Due to differences in $B$, the apparent surface echo observed by each radar system probes the near-surface to different depths, and the associated $\varepsilon_{eff}$ is

characteristic of the near-surface properties spanning that depth.

| Radar System | $f_c$ (MHz) | $B$ (MHz) | $z_{0,firn}$ (m) | $z_{0,ice}$ (m) |
| --- | --- | --- | --- | --- |
| HiCARS2 & MARFA | 60 | 15 | 11.3 | 8.5 |
| MCoRDS3 | 195 | 30 | 5.7 | 4.3 |

Table 1. Near-surface depth (i.e., range resolution $z_0$) calculated with Eq. (1) for airborne ice-penetrating radars used in this study, representative of a medium composed entirely of firn or ice. For firn, $z_{0,firn}$ was calculated with a relative permittivity of $\varepsilon = 1.8$, equivalent to a density of $\rho = 417$ kg m$^{-3}$. $f_c$ is the center frequency, and $B$ is the bandwidth.


We apply the RSR method to all airborne IPR datasets in this study. In the RSR technique, a Homodyne K-distribution is fit to surface radar amplitudes over a specified along-track window, to deconvolve the signal into its coherent/specular ($P_c$) and incoherent/scattered ($P_n$) components (Grima et al., 2014a). Surface power was corrected for geometric spreading loss to account for variations in aircraft altitude. Different bin sizes were used to produce the RSR-derived products of the

HiCARS2/MARFA and MCoRDS3 systems to account for differences in sampling rates along-track. For MCoRDS3, a

window of 5000 samples (equivalent to ~1.5 km) with a step size of 1250 samples was used, providing about 75% overlap. For both HiCARS2 and MARFA, a window of 1000 samples (equivalent to ~1 km) with a step size of 250 samples was used, also providing about 75% overlap.

$P_c$ is mainly sensitive to permittivity changes, which is governed by the near-surface composition and structure (e.g., layers). On the other hand, $P_n$ is mainly sensitive to the surface roughness and random non-stratigraphic heterogeneities (e.g., voids) within the near-surface. In this work, we focus on $P_c$ to constrain the thickness of ice layers in Zone II of the DIC firn. $P_c$ can be further described as (Ulaby et al., 1981)

$$P_c = r^2 e^{-\left(\frac{4\pi\sigma_h}{\lambda}\right)^2},$$    (2)

where $r$ is the effective surface reflection coefficient in amplitude, $\sigma_h$ is the root mean square (RMS) height over the wavelength scale, and $\lambda$ is the wavelength. Although $P_c$ is a function of the RMS height, Rutishauser et al. (2016) found that surface roughness is not the main contributor to surface scattering over DIC. The mean surface roughness derived from laser
altimetry observations over DIC in Zone II is $\sigma_h = 0.09$ m (Rutishauser et al., 2016). Based on this value, we quantify the effects of surface roughness on $P_c$, represented by the exponential term of Eq. (2). We find that surface roughness contributes 0.22 dB to $P_c$ (i.e., approximately 1% in terms of dB). Although surface roughness is dependent on ice cap conditions during data acquisition (e.g., the presence of freshly fallen snow), we assume $\sigma_h = 0.09$ m remains representative of surface roughness at the time the SRH1 survey was conducted, especially given the minor contribution of surface roughness to $P_c$.
Both GOG3 and SRH1 datasets were collected in the spring prior to the onset of the melt season. Thus, surface roughness would be governed by the presence of snow (i.e., appears smoother to airborne radar), which we assume to be the case for both surveys. We also note that 0.22 dB represents a highly conservative value, because $\sigma_h$ values derived from laser altimetry in Rutishauser et al. (2016) considered a baseline of ~171.5 m along-track. The surface roughness at the wavelength scale ($\lambda = 5$ m) is expected to be smaller, because surface roughness scales with the baseline of interest (Shepard
et al., 2001).

Given that surface roughness contributes marginally to $P_c$, we hypothesize that $P_c$ will be predominantly sensitive to $r^2$ as opposed to surface roughness. The DIC firn provides the opportunity to study how $P_c$ varies with changes in the coherent near-surface structure/geometry through the various interferences between the reflections arising from the dielectric
interfaces making up the firn/ice stack.

For this work, we focus on relative variations in coherent power between all surveys; thus, absolute calibration of the signal is not necessary, although coherent power from the HiCARS2 survey was previously calibrated as noted in Rutishauser et al.

(2016). Additionally, to ensure the underlying assumptions for RSR are met, data with a correlation coefficient (a goodness-of-fit estimator for the RSR technique) below 95% and an aircraft roll above 2.9° are excluded from the analysis. Previous applications of the RSR method have empirically shown that an aircraft roll of 2 to 3° allows for a stable coherent radar return. This roll is consistent with a half beamwidth at nadir to maintain signal stability to +/- 1 dB (Peters et al., 2007). Data at surface elevations below 800 m are also excluded to remove observations collected over rock outcrops.

## 2.2 Surface-based radar and firn cores

We use surface-based radar data and firn cores to serve as validation and ground-truth for interpreting the airborne radar observations over DIC. Surface-based radar surveys were conducted in spring 2015 along several HiCARS2 transects (Fig. 1). The surface-based radar system consisted of a PulseEKKO Noggin radar with a center frequency of 500 MHz, collecting data sampled at ~0.4 m along-track (Rutishauser et al., 2016). We use the surface-based radar data along the profile marked in Fig. 1 that traverses Zones II and III. Here, we identify an ice slab in the firn by picking the firn-ice interface, and its thickness is used as an input to a thin layer reflectivity model (see Sec. 2.3). The ice slab was obtained from manually picking the surface-based radargram (Fig. 4b). Although the top of the ice slab is easily distinguishable in the radargram, the bottom of the ice slab is highly uncertain in several locations. Therefore, the bottom of the ice slab was picked as the first continuous reflection indicating an ice-firn transition as best as possible. Firn cores were collected along the surface-based radar profiles during the same survey in spring 2015 (Fig. 1) (Rutishauser et al., 2016). Sections of the firn cores were used to derive the background/dry firn density (where no ice layers were present) at various depths (Fig. S1). The average density measurement error is estimated to be around 50 kg m$^{-3}$; however, we observe a few obvious outliers that we do not use in our analysis (Fig. S1).

## 2.3 Model description

The observed surface power from airborne IPR is the sum of all electric fields reflected and scattered due to dielectric contrasts within the range resolution and bounded spatially by the pulse-limited footprint (Grima et al., 2014a). As such, the observed power is expected to be modulated by near-surface properties. Following previous studies, we model the reflection of electromagnetic radiation from a medium consisting of porous and fully densified ice layers (Mouginot et al., 2009; Grima et al., 2014b). The model is based on the transfer-matrix implementation, describing the interaction of an electromagnetic wave with a stratified medium consisting of homogeneous layers (Born and Wolf, 1970). More specifically, individual propagation matrices are generated with an assigned relative permittivity for each layer and an assigned thickness for all but the uppermost and bottommost layers in the stack. The product of these propagation matrices forms the characteristic matrix for the entire stack. The reflection coefficient $r^2$ for the medium can then be obtained from this characteristic matrix.

We model the reflectivity of stratified layers over DIC from HiCARS2, using both 3-layer and 4-layer stack configurations (Fig. 4c). In this work, we refer to the porous ice layer above and below the ice slab as the firn$_1$ and firn$_2$ layer (when

applicable), respectively. Values obtained from in-situ measured firn densities (Fig. S1) were converted to permittivities using an empirical model relating firn/ice density ($\rho$ [kg m$^{-3}$]) to relative permittivity ($\varepsilon$) (Kovacs et al., 1995), calculated as

$$\varepsilon = (1 + 0.000845\rho)^2 . \tag{3}$$


Using Eq. (3), $\varepsilon = 1.8$ was obtained from $\rho = 417$ kg m$^{-3} \pm 40$ kg m$^{-3}$ for firn$_1$, by averaging firn core density measurements taken from the surface to 1 m depth. Similarly, $\varepsilon = 2.2$ was obtained from $\rho = 584$ kg m$^{-3} \pm 56$ kg m$^{-3}$ for firn$_2$, by averaging firn core density measurements taken between 2.5 and 11 m depths, excluding the two outliers with densities greater than that of glacier ice (Fig. S1). $\varepsilon = 3.15$ was assigned to the ice slab (Fujita et al., 2000). For simplicity, we assign these bulk

densities/permittivities to the firn layers in our model but acknowledge that the firn density is expected to vary horizontally and vertically over the DIC near-surface.

The model does not account for surface roughness, of which effects on $P_c$ are assumed to be negligible over DIC. The model also does not account for relatively small-scale heterogeneities, such as residual firn embedded within a fully densified ice

layer, and ice lenses embedded in the firn layers. We discuss the implications of these assumptions in the following section.

The HiCARS2 chirp is not monochromatic. Therefore, to consider the effects of all frequencies defined by the chirp's center frequency and bandwidth, we model the reflectivity from a particular stack configuration ($R_N$) as (Mouginot et al., 2009)

$$R_N = max(||IFFT\big(S(f)r(f, \delta z_{1…N}, \varepsilon_{1…N})S^*(f)\big)||^2) , \tag{4}$$

where $f$ is the frequency, $S$ is the linearly modulated chirp in the frequency domain, $N$ denotes the number of layers considered in the model, $\delta z$ is the thickness of a particular layer, and $\varepsilon$ is the relative permittivity of that layer in the stack configuration. IFFT is the inverse fast Fourier transform, the operator $|| … ||$ denotes the magnitude, and the asterisk indicates

the complex conjugation of the chirp. A synthetic chirp was generated to represent $S$ for HiCARS2 in this study, using a pulse length of 1 µs, 50 MHz sampling rate, and 1200 samples. The same sampling interval for $S$ and $r$ were used over the system bandwidth.

# 3 Results

## 3.1 Dual-frequency characterization of near-surface structure


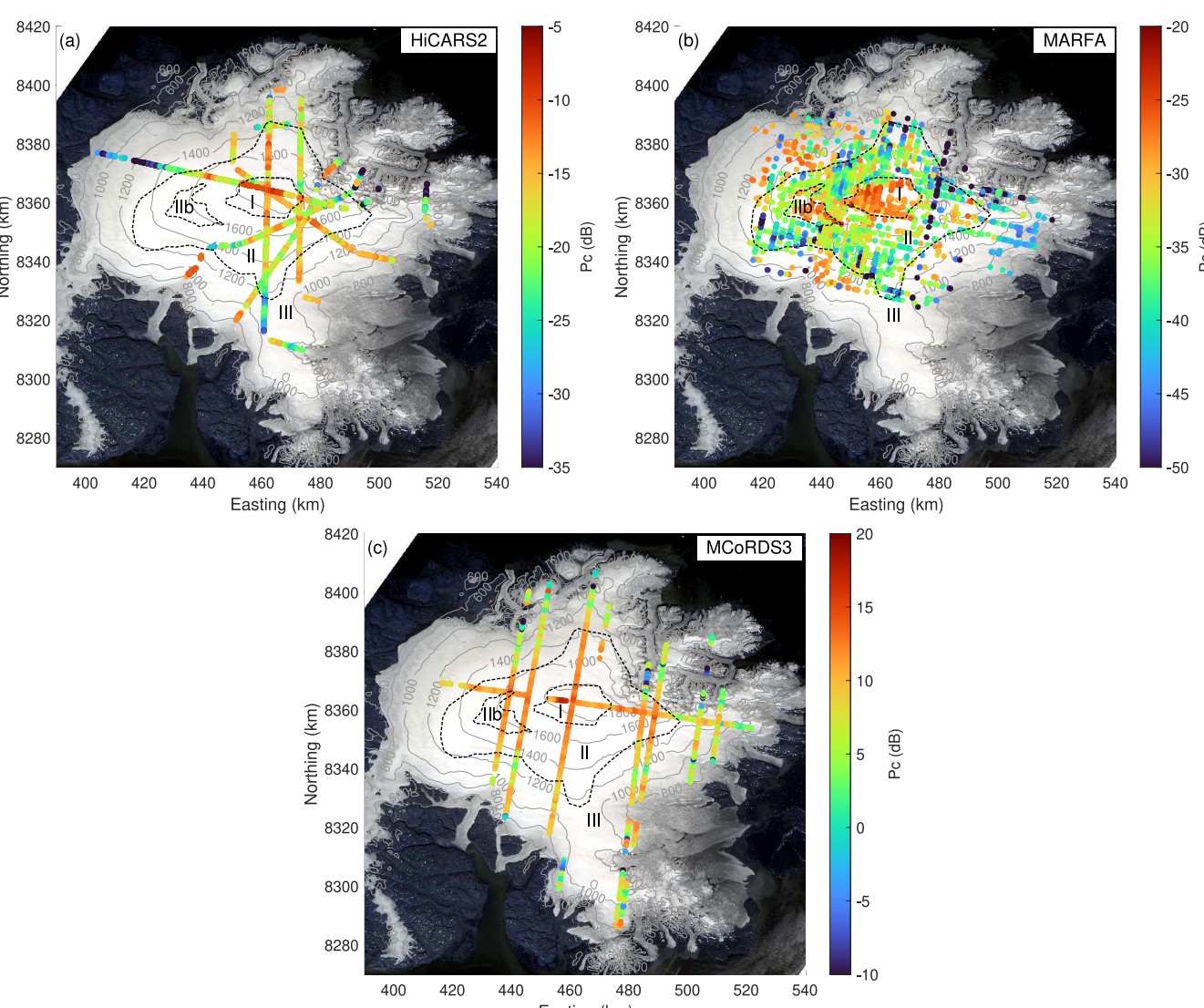

**Figure 2. Maps of surface coherent power ($P_c$) over Devon Ice Cap derived from radar data collected with (a) HiCARS2, (b) MARFA and (c) MCoRDS3, all scaled to the same dynamic range to facilitate comparison between the surveys. Background, elevation contours, and zone boundaries as in Fig. 1.**

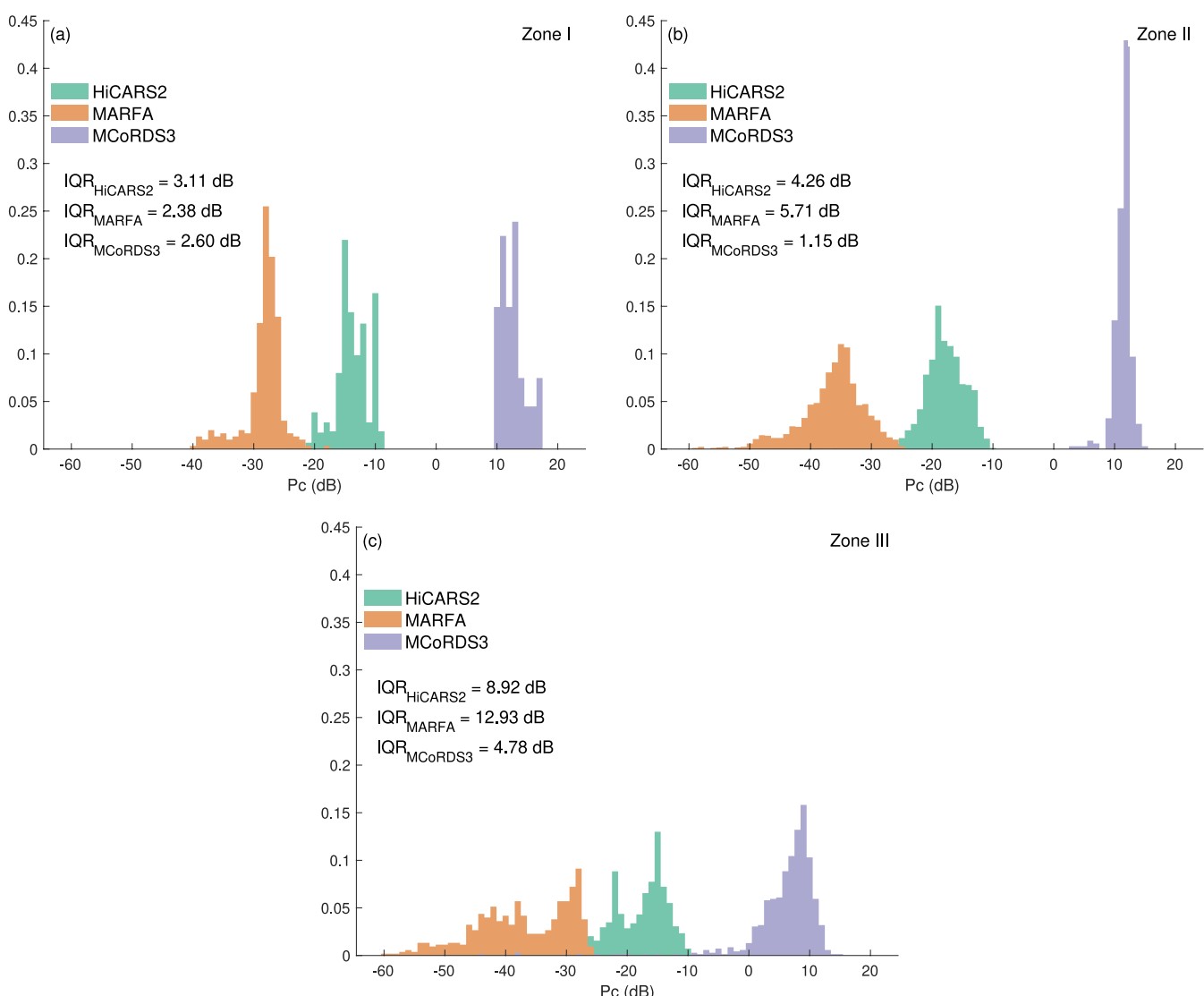

**Figure 3. Normalized histogram distributions of surface coherent power ($P_c$) over Devon Ice Cap across (a) Zone I, (b) Zone II and (c) Zone III, by airborne ice-penetrating radar survey and their corresponding interquartile ranges (IQR).**

Differences in range resolution between two airborne radar systems, operating at different frequencies and bandwidths, can be used to investigate bulk properties of the near-surface (Chan, 2022). This approach is useful in the absence of data capable of resolving the near-surface stratigraphy (e.g., firn cores, surface-based radar, or high-frequency airborne ice-penetrating radar). Here, we analyze $P_c$ derived from HiCARS2/MARFA and MCoRDS3 to evaluate this approach (Fig. 2). We calculate the interquartile range (IQR) of $P_c$ distributions for each survey as a measure of $P_c$ variability within each zone (Fig. 3). We then compare $P_c$ derived from HiCARS2 to modeled reflectivity along a transect, to assess the influence of different firn properties on $P_c$ in a layered medium. Results from our model, with input from surface-based radar data and firn cores, provide validation for our interpretation of the dual-frequency airborne IPR datasets.

### 3.1.1 Updated firn zone boundaries

Prior to conducting the dual-frequency analysis, firn zone boundaries previously identified in Rutishauser et al. (2016) were updated using a similar method with the more recent MARFA survey. Rutishauser et al. (2016) previously hypothesized that ice slabs in firn cause increased scattering of the radar return relative to thin ice lenses or glacial ice. We use the $P_c/P_n$ ratio as an indicator of relative scattering from the near-surface that excludes the effects of permittivity (Grima et al., 2014a). Firn zone boundaries were refined based on visual inspection of the spatial distribution of the $P_c/P_n$ ratio [dB]. These ratio values were obtained by taking the difference between $P_c$ and $P_n$ expressed in decibels. Zone II consists of ratio values less than 0 dB (i.e., $P_n$ dominating), whereas Zones I and III consist of ratio values greater than 0 dB (i.e., $P_c$ dominating) (Fig. S2). The largest discrepancy between the old and new boundaries occurs in the northwest part of DIC, where the new Zone II/III boundary has migrated to higher elevations. When compared with Landsat-8 images taken in August 2019 (late into the melt season), we observe that the new Zone II/III boundary is in good agreement with a transition from snow/firn to exposed bare ice/meltwater (Fig. 5). The images validate the lack of firn at the surface in the northwest part of DIC. We find that this method for delineating firn zones is valid and advantageous, because it is also sensitive to bulk changes in the firn stratigraphy between Zones I and II otherwise invisible to optical imagery.

### 3.1.2 Zone I: thin ice lenses in firn

Visual inspection of $P_c$ observed across all three surveys within Zone I indicate their overall similar character (Fig. 2). IQRs within Zone I are low, indicative of relatively less variability in $P_c$ (Fig. 3). Interzonally, consistently high $P_c$ values are also observed for all three airborne IPRs in Zone I compared to the other two zones by airborne IPR survey. Altogether, these observations suggest the bulk firn structure observed in Zone I appears similar to all airborne IPRs, spatially and vertically, to depths bounded by the range resolutions of each system. The firn in Zone I is known to host thin, flat ice lenses (Rutishauser et al., 2016), which can generate coherent radar reflections. Firn cores collected in Zone I confirm the existence of ice lenses from the surface to ~11 m, which is also the depth approximately equivalent to the theoretical range resolution of HiCARS2/MARFA (Table 1). While MCoRDS3 is expected to be more sensitive to ice lenses than HiCARS2/MARFA, owing to its higher frequency, the integration of coherent reflections from multiple ice lenses at depths within the range resolution of each airborne IPR could be responsible for the high $P_c$ values observed in Zone I across all three surveys.

### 3.1.3 Zone II: thick ice slabs in firn

$P_c$ values from MCoRDS3 appear stable across the Zone I/II boundary (Fig. 2). However, a change in the pattern of $P_c$ is observed for HiCARS2/MARFA between Zones I and II (Fig. 2), indicative of a change in the bulk near-surface firn structure that primarily affects the 60 MHz airborne IPR's coherent response. Within Zone II, IQRs of $P_c$ from both HiCARS2 and MARFA are more than three times greater than MCoRDS3 (Fig. 3b). HiCARS2 and MARFA operate with the same range resolution and thus are expected to observe the firn column to the same depth, despite the surveys being

conducted in different years. MCoRDS3 operates with a finer range resolution, and thus the surface signal probes the firn to shallower depths.

In Zone II, we attribute differences in $P_c$ variability between HiCARS2/MARFA and MCoRDS3 to changes in the near-surface firn properties to different depths, limited by each airborne IPR's range resolution. Zone II is known to host meters-thick ice slabs in the top 10 m of the firn column, in addition to thin ice lenses embedded in the firn (Gascon et al., 2013a; Rutishauser et al., 2016). The larger range resolution of HiCARS2/MARFA indicates that $P_c$ is affected by features in the firn column at depths beyond the range resolution of MCoRDS3, with the main feature being the bottom ice slab interface.

Additional heterogeneities at depth, such as interstitial firn or variations in the ice slab thickness, could also contribute to $P_c$ variability. The integration of reflections from these heterogeneities within the HiCARS2/MARFA range resolution could drive large fluctuations in $P_c$ behavior due to constructive/destructive interferences. This would explain the higher IQRs observed in Zone II by HiCARS2/MARFA compared to MCoRDS3. However, we note a region Zone IIb within Zone II (Fig. 2b), where anomalously strong $P_c$ values from MARFA are observed. $P_c$ within Zone IIb behaves similarly to $P_c$ in

Zones I and III, suggesting the presence of either relatively homogeneous firn or fully densified glacier ice. Zone IIb could alternatively feature interstitial firn in an ice slab resulting in higher $P_c$ values relative to its surroundings, as similarly observed from 25 to 28 km along the surface-based radargram (Fig. 4). Additionally, the $P_c/P_n$ ratio approximately equals 0 dB in Zone IIb and appears distinct from the other three zones (Fig. S2). This subzonal region may represent a new firn zone with inclusions that equally scatters and reflects the radar return but less heterogeneous than the surrounding Zone II.


The finer range resolution of MCoRDS3 limits $P_c$ sensitivity to the firn$_1$ layer and partially the ice slab. Any changes in firn structure at depths beyond the MCoRDS3 range resolution are not captured by $P_c$. We interpret $P_c$ from MCoRDS3 in Zone II to be sensitive to layers consisting primarily of thin ice lenses embedded within the firn$_1$ layer and the top ice slab interface, all expected to generate coherent reflections. The depth of the top ice slab interface changes little relative to the

bottom interface (Rutishauser et al., 2016). Due to limited firn heterogeneities and variations in firn structure within the range resolution of MCoRDS3, there is less observed variability in $P_c$ from MCoRDS3 in Zone II. Thus, the Zone II near-surface appears spatially and vertically homogeneous to MCoRDS3, consistent with the similar pattern of high $P_c$ values in Zone I.

### 3.1.4 Zone III: fully densified glacier ice

The pattern of $P_c$ from MCoRDS3 is noticeably different in Zone III compared to that of Zones I and II (Fig. 2c), where $P_c$ from MCoRDS3 is generally lower in Zone III (<10 dB) compared to Zones I and II (≥10 dB). Where this pattern shift occurs coincides with the Zone II/Zone III boundary defined using only MARFA data, thus further validating this approach for delineating zones. Interzonal comparison of $P_c$ by airborne IPR survey indicates greater variability in Zone III compared

to the other two zones (Fig. 3), suggesting another change in the near-surface structure of Zone III that likely affects the return from all airborne IPRs. Intrazonally, IQRs within Zone III are dissimilar when compared to each other, which further suggests that the near-surface structure affects each airborne IPR's surface return in different ways. Zone III consists mainly of fully densified glacier ice without firn and theoretically should produce a strong and stable coherent return. The fact that we do not observe this behavior throughout all of Zone III indicates the near-surface structure appears spatially and vertically heterogeneous for all three airborne IPRs.

While $P_c$ is challenging to interpret in Zone III, the dual-frequency airborne IPR approach can provide insight to possibly explain these observed variations. For example, we investigate how the snow cover alone in Zone III, prior to the onset of the melt season, might influence $P_c$ at the two frequencies of interest. Spring snow depth measurements on DIC indicate the snow layer thickness can vary tens of centimeters, increasing from northwest (<50 cm) to southeast (>100 cm) across the ice cap (Koerner, 1966). To study the effects of such a layer, we model its reflectivity embedded in between semi-infinite half spaces of air above and fully densified ice below, as a function of snow layer thickness and $\varepsilon$ (Fig. S3). For typical snow densities (equivalent to $\varepsilon \leq 1.7$), reflectivity is found to only vary by several dB for most snow layer thicknesses. Exceptions occur when $\varepsilon$ approaches ~1.7 or fairly dense snow at specific layer thicknesses (Fig. S3). To generate the $P_c$ variability observed by both frequencies in Zone III, a ~90 cm thick snow cover with centimeter scale thickness variability and $\varepsilon \approx 1.7$ would be required throughout this zone. This scenario is inconsistent with spring snowpack thicknesses across DIC (Koerner, 1966) and thus cannot fully account for $P_c$ variations in Zone III.

An alternative hypothesis may involve remnant supraglacial river channels carved during the previous melt season (see Sec. 4.1). These channelized features vary spatially across Zone III (Fig. 5) and may retain their structure when refrozen and buried underneath the winter snowpack. These snow-filled, lineated structures may induce non-negligible effects on $P_c$, as a result of surface roughness and snow depth variability. Further analysis to test this hypothesis, while potentially insightful, remains out of scope for this work.

### 3.2 Coherent power sensitivity to heterogenous near-surface structure: validation and ground-truth

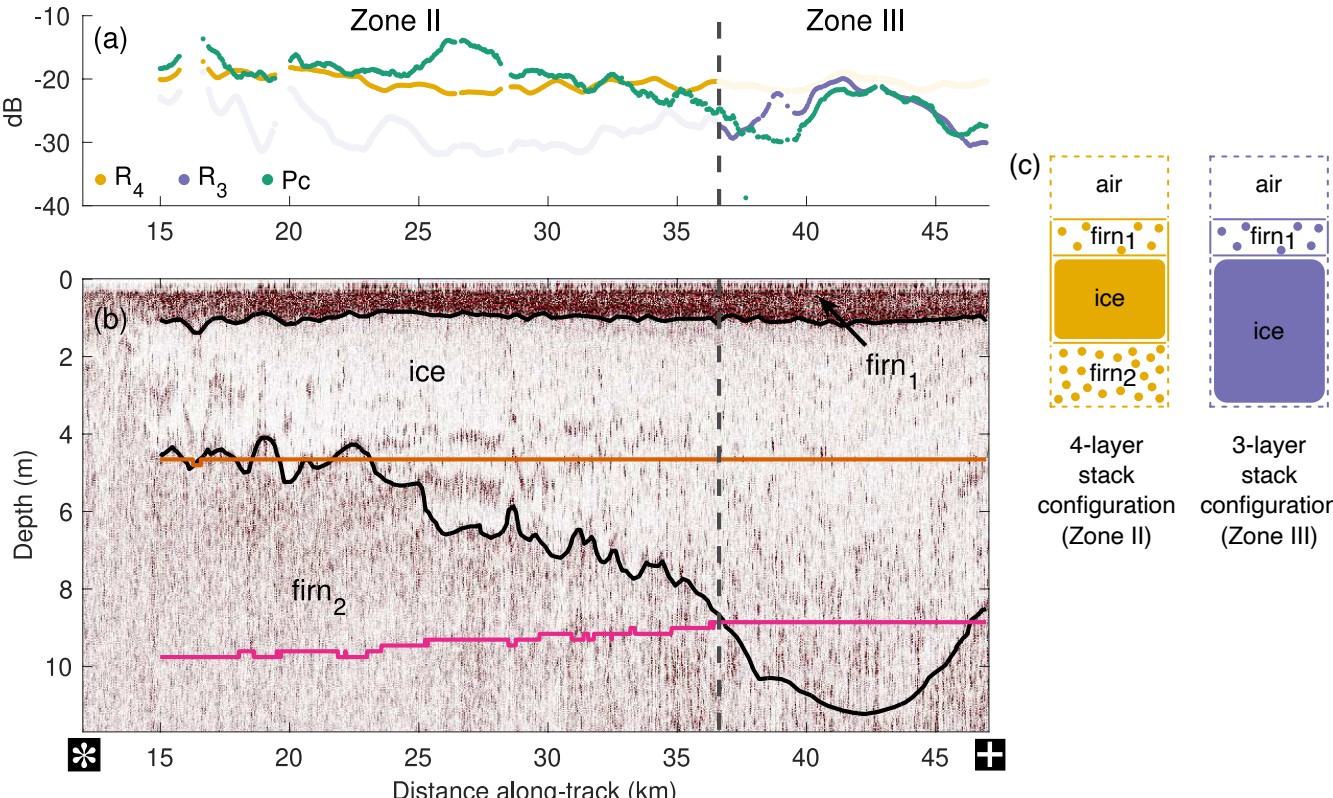

Figure 4. (a) HiCARS2 surface coherent power ($P_c$) along the segment from '*' to '+' of transect NDEVON/JKB2k/Y4a in Fig. 1 and modeled reflectivity from 3-layer ($R_3$) and 4-layer ($R_4$) stack configurations. (b) Surface-based radargram along the same segment, adapted from Rutishauser et al. (2016), with the Zone II to III boundary marked by the vertical black dashed line. Ice-firn transitions were manually picked along this profile (black solid lines). Range resolutions of the surface return (i.e., $z_0$) are depicted for HiCARS2/MARFA (pink solid line) and MCoRDS3 (orange solid line), based on the assumed layer thicknesses and densities of the firn/ice stack along this profile. (c) Graphical representation of the 3-layer and 4-layer stack configurations used in the reflectivity model.

To validate our interpretation of the airborne IPR datasets by assessing how the vertical firn structure over DIC influences $P_c$, we compare modeled reflectivity $R_N$ (where $N$ indicates the stack configuration) to previously absolutely calibrated $P_c$ derived from HiCARS2 (Rutishauser et al., 2016) along a transect with collocated surface-based radar observations (Fig. 4). Between the two stack configurations represented in our model (Fig. 4c), there is better agreement between $R_4$ (4-layer stack) and $P_c$ from about 15 to 35 km along-track in Zone II. In contrast, $R_3$ (3-layer stack) better approximates $P_c$ from about 37 to 47 km along-track in Zone III. The shift from $R_4$ to $R_3$ occurs at the Zone II/III boundary, where the picked bottom ice slab interface (i.e., the ice-firn2 interface) lies ~9 m beneath the surface (Fig. 4b). This depth coincides with the theoretical range resolution (Table 1) and thus also represents the apparent range resolution of HiCARS2. While our model does not take into account range resolution constraints and integrates all coherent reflections picked from the surface-based radar data, we find that it adequately predicts $P_c$ along the profile best described by a stack configuration characteristic of the

near-surface stratigraphy. Only near-surface coherent reflections generated within the range resolution contribute to $P_c$. This implies that the reflection from the ice-firn$_2$ interface is integrated into $P_c$ in Zone II but not in Zone III. Our interpretation of $P_c$ is consistent with the transition from embedded ice layers in the firn of Zone II to the fully densified glacier ice of Zone III (Gascon et al., 2013a; Rutishauser et al., 2016). Accordingly, the lower ice slab interface was picked to reside at depths well below the expected HiCARS2 range resolution in Zone III (Fig. 4b).

Consideration of uncertainties in our model from measured firn densities yield reflectivity variations ranging from 1.5 to 2.9 dB, which can explain the slight offset between $R_N$ and $P_c$ but are still smaller than the full range of $P_c$ values observed (~15 dB) along this profile. Differences between $R_N$ and $P_c$ could also be attributed to smaller inhomogeneities within the ice slab and firn layers. For example, interstitial firn can be seen within the ice slab from 15 to 18 km and from 25 to 28 km along the profile (Fig. 4b) but is not represented in our model. Such layers can likely generate additional reflections that could constructively interfere and thus explain why $P_c$ exceeds $R_4$ in those parts of the profile. We also do not account for ice lenses (cm-scale) in the firn$_1$ and firn$_2$ layers. Modeling of HiCARS2 surface reflectivity from an ice lens in firn accounts for <1 dB relative to a firn column without an ice lens (Fig. S4), although the presence of multiple ice lenses may have non-negligible effects. Uncertainties due to our manual picks were considered and shown in Fig. S5, by varying the pick of the lower ice slab interface. Results indicate minimal changes to modeled reflectivity values and cannot resolve the discrepancies between $R_4$ and $P_c$, which further indicates our model is too simplistic to account for smaller complexities (i.e., multiple ice-firn transitions) that arise from heterogeneous melting and refreezing processes in firn. In light of the uncertainties responsible for mismatches between the modeled and observed values, our results indicate that the near-surface heterogeneous structure featuring an ice slab governs $P_c$ from HiCARS2 to a first-order, when coherent reflections are generated within the HiCARS2 range resolution. This is also consistent with our interpretation of the dual-frequency airborne IPR datasets over DIC.

## 4 Discussion

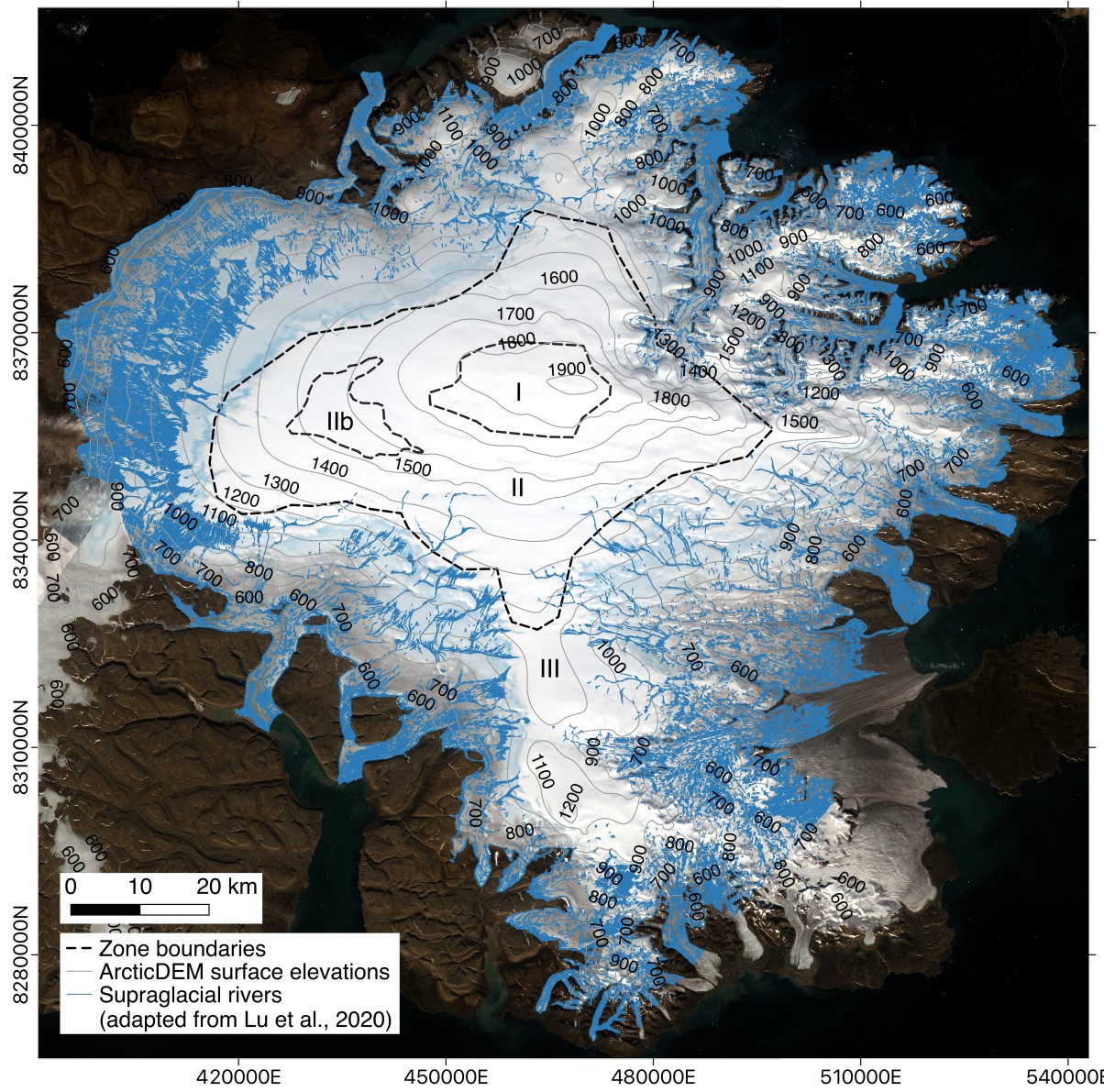

**Figure 5. Landsat-8 images (courtesy of the U.S. Geological Survey) of Devon Ice Cap taken mid-August 2019, overlain with zone boundaries derived from MARFA and supraglacial rivers previously mapped by Lu et al. (2020). Thin gray lines indicate elevation contours displayed from 600 m to 1900 m every 100 m.**

### 4.1 Near-surface hydrological conditions of Devon Ice Cap

Our results along with those of Lu et al. (2020) indicate the DIC near-surface firn may exhibit conditions that favor lateral runoff over vertical infiltration of surface meltwater in certain regions across the ice cap. The thickness of ice slabs within

Zone II affects the bulk firn permeability and thus the capability of vertical meltwater infiltration. Based on differences between the range resolutions of airborne IPRs in this study, we can estimate the average thickness of ice slabs in Zone II. Because MCoRDS3 is expected to only capture the upper interface of the ice slab while HiCARS2/MARFA captures its

entirety, the difference between their range resolutions provides an estimate of average ice slab thickness ($h$), calculated as

$$h \approx z_{0,mix}^{HiCARS2/MARFA} - z_{0,mix}^{MCoRDS3}, \tag{5}$$

where $z_{0,mix}^{IPR}$ is the range resolution of an airborne IPR system associated with a heterogeneous medium/mixture of $\varepsilon_{eff}$

defined by Eq. (1). While usually unknown, $\varepsilon_{eff}$ for a mixture composed of firn and ice (e.g., an ice slab in firn) is expected to be bounded by the permittivity of their end member components. Based on Eq. (1), $z_{0,mix}^{IPR}$ is then also expected to be bounded by the range resolutions defined for a homogeneous medium of either firn ($z_{0,firn}^{IPR}$) or ice ($z_{0,ice}^{IPR}$). This implies that the difference between the range resolutions of HiCARS2/MARFA and MCoRDS3 when probing a homogeneous medium of firn or ice yields a range of average ice slab thicknesses, such that $(z_{0,ice}^{HiCARS2/MARFA} - z_{0,ice}^{MCoRDS3}) < h <$

$(z_{0,firn}^{HiCARS2/MARFA} - z_{0,firn}^{MCoRDS3})$. From this, we estimate average ice slab thicknesses range from 4.2 to 5.6 m in Zone II on DIC, based on values provided in Table 1. These estimates are consistent with surface-based radar profiles, particularly at lower elevations within Zone II (Gascon et al., 2013a; Rutishauser et al., 2016).

Our ice slab thicknesses are thicker than the impermeable thickness threshold of at least 1 m noted by Ashmore et al. (2020)

and overall similar to the observed thickness of Greenland's ice slabs, which can limit or substantially delay meltwater percolation (Charalampidis et al., 2016; Machguth et al., 2016; MacFerrin et al., 2019). Spatially, the radar response associated with refrozen ice slabs in Zone II is widespread (Fig. 2), suggesting impermeable ice slabs are most likely ubiquitous and continuous in Zone II except in the anomalous subzonal region, Zone IIb. Here, discontinuous thin ice lenses could be present instead, similar to Zone I as opposed to the fully densified glacier ice of Zone III, since firn is present at

these elevations (Fig. 5). Additional data (e.g., firn cores and surface-based radar) would help characterize the firn properties of Zone IIb.

Given the hypothesized thickness and continuity of ice slabs in DIC, we find that the near-surface structure of Zone II could enable lateral meltwater runoff atop ice slabs through the firn$_1$ layer. However, it is uncertain as to whether some of this

meltwater would refreeze within the firn during lateral runoff. Any latent heat released from refreezing would raise firn temperatures and promote subsequent meltwater runoff, due to the lack of sufficient cold content that would otherwise facilitate refreezing (Vandecrux et al., 2020a). The firn$_1$ layer is relatively thin (~1 m) and can experience seasonal temperature fluctuations through conduction with the atmosphere. Therefore, during the melt season, atmospheric warming could also raise firn temperatures and promote lateral runoff. The extent to which runoff occurs depends on the balance

between the volume of surface meltwater produced and the available firn storage capacity in each basin, which is modulated by the thermal and hydraulic properties of the firn that control infiltration and refreezing.

In Zone III, Lu et al. (2020) mapped seasonal supraglacial river networks on DIC from imagery collected in 2016, where some rivers appear to form in Zone II (Fig. 5). This suggests that the firn stratigraphy likely influences the development of
supraglacial meltwater channels. We hypothesize that high-elevation surface meltwater runoff over ice slabs in Zone II may contribute to the total meltwater supply routed through supraglacial rivers in the ablation zone. If enough meltwater is produced and saturates the firn, then slush flows could also develop (Fernandes et al., 2018; Pitcher and Smith, 2019), particularly near the Zone II/III boundary. The development of both dendritic and parallel supraglacial river networks observed on DIC (Lu et al., 2020) might imply how meltwater is routed through preferred pathways through the overlying
firn. Future studies of the Zone II firn structure may reveal how it could modulate water fluxes at the accumulation/ablation zone boundary that could drive some of the observed but unexplained variations in the channel structure in the upper ablation zone. Because the Zone II/III boundary tracks the visible runoff line depicted by Landsat-8 imagery (Fig. 5), repeat airborne IPR surveys of DIC would capture shifts in the Zone II/III boundary over time that may correspond to spatial shifts in physical properties controlling runoff.


A lack of supraglacial rivers extending upslope into Zone II is observed in northwest DIC (Fig. 5). This could be attributed to where rivers had not fully developed during the melt season when mapped by Lu et al. (2020). While the full mapping of supraglacial rivers used images taken in late July, Lu et al. (2020) also found that rivers in certain regions extend to higher elevations later into the melt season. This includes western DIC, where rivers develop at higher elevations between 1400 to
1500 m in mid-August (Lu et al., 2020), coinciding with the Zone II/III boundary and corresponding to when the Landsat-8 images were captured (Fig. 5). The relatively shallow slopes of northwest DIC may influence the rate at which rivers form compared to other regions across the ice cap. In addition, the 10 m resolution of images used by Lu et al. (2020) could have prevented detection of relatively narrow streams that otherwise are present.

### 4.2 Dual-frequency/bandwidth radar reflectometry: advantages and limitations

Dual-frequency/bandwidth reflectometry on its own can be used to garner insights into the near-surface heterogeneity of firn (i.e., presence of ice slabs in this study), especially when lacking prior knowledge of the general firn structure. If we consider the case of homogeneous firn, $P_c$ is assumed to be mainly sensitive to surface density variations (Grima et al., 2014b). In this case, the radar response is non-dispersive (independent of frequency), and $P_c$ derived from both frequencies/bandwidths would appear to be the same in a dry firn column without ice slabs. However, a dual-frequency system can confirm (to
depths constrained by the bandwidth-limited range resolution) whether $P_c$ is truly representative of surface density or instead, affected by ice slabs if present. In the latter case, we would expect a difference in $P_c$ between both radar systems, because $P_c$ is affected by layer density and thickness. We caution against the inversion of $P_c$ for surface density similar to previous

applications of RSR (Grima et al., 2014b, 2016), because inverting $P_c$ from mono-frequency airborne IPR data for surface density can be ambiguous in regions dominated by significant surface melting and refreezing. Future studies utilizing dual-frequency/bandwidth reflectometry could help validate results from multidimensional firn models to better capture complex heterogeneous meltwater percolation and refreezing in firn (Vandecrux et al., 2020b).

While the $P_c/P_n$ ratio was used to define the spatial extent of ice slabs independent of surface-based radar data (see Sec. 3.1.1), the Zone II/III boundary also constrains the maximum ice slab thickness over DIC. Ice slabs grow in thickness from higher to lower elevations and are not expected to exist beyond the Zone II/III boundary due to the lack of firn in Zone III. However, applying a similar approach to other regions may lead to ambiguous results. For example, ice slabs in Greenland have been found to be thicker than ones on DIC (MacFerrin et al., 2019). If an ice slab continues with depth beyond the range resolution of both radar systems, then $P_c$ from dual-frequency/bandwidth reflectometry would be unable to distinguish between a thick ice slab (with underlying firn) and fully densified glacier ice. In this case, our approach would yield a minimum average ice slab thickness, because the surface return is only characteristic of bulk firn properties within the limits of the range resolution. Incorporating additional knowledge of the region, such as the location of the firn line, would be helpful for constraining both the spatial distribution and thickness of ice slabs.

Our study also demonstrates quantitatively the first proof of concept for using a dual-frequency approach to characterize the near-surface of extraterrestrial environments. Such an approach is applicable to where dual-frequency spaceborne IPR data already exists, such as on Mars, or will be collected at Jupiter's moon Europa. Future exploration with the dual-frequency radar sounder onboard NASA's Europa Clipper mission can apply a similar approach to detect and characterize near-surface ice layers and overburden. Unlike on Earth, ice layers may form from hypothesized brine infiltration and refreeze in icy regolith on Europa (Chan et al., 2017; Schmidt et al., 2011), thus identifying locations of potential near-surface endogenic activity.

**5 Conclusion**

We present the first study aimed to characterize the vertical and spatial extent of ice layers in firn over an entire ice cap with dual-frequency airborne IPR. We demonstrate how each system captures near-surface heterogeneity to different vertical extents, limited by their bandwidth-constrained range resolutions. Our results indicate that the surface coherent power derived from RSR is sensitive to bulk firn properties featuring ice layers embedded in firn. This is supported by modeled reflectivity values that visually correlate well with the observed coherent power on DIC, based on the stack configuration that best describes the near-surface structure of the corresponding zone. We further leverage the differences in range resolution to constrain the thickness of ice slabs within Zone II. Our thickness estimates imply that ice slabs are impermeable and thus capable of impeding vertical meltwater percolation in favor of lateral runoff in Zone II. Ice slabs are likely

pervasive, which suggests that lateral flow may dominate over deep percolation and local retention in Zone II and feed supraglacial rivers downslope. Coupled with warming temperatures (Mortimer et al., 2016), we predict that the DIC near-surface will continue to promote surface meltwater runoff if such conditions are sustained.

## Data Availability

The surface-based radar and firn core data are available at https://doi.org/10.5281/zenodo.7544347. The derived RSR products for all airborne IPR surveys used in this study are available at https://doi.org/10.18738/T8/QKGFGX.

## Author Contributions

KC, CG, AR, and DDB conceptualized the study. KC developed the model, led the data visualization, and wrote the manuscript. AR collected and processed the surface-based radar and firn core data. DAY processed and curated the HiCARS2 and MARFA datasets. CG derived the RSR products for the HiCARS2 and MARFA datasets. RC derived the RSR products for the MCoRDS3 dataset. KC, CG, AR, RC, and DDB contributed to data analysis and interpretation. DDB acquired funding for this study. All authors reviewed and edited the manuscript.

## Competing Interests

The authors declare that they have no conflict of interest.

## Acknowledgements

We acknowledge the use of the surface-based radar data and firn cores collected with support from NSERC (Discovery Grant/Northern Research Supplement), Alberta Innovates Technology Futures, the CRYSYS Program (Environment Canada), and a University of Alberta Northern Research Award. The use of the GOG3/HiCARS2 radar data was funded by UK NERC grants NE/K004999/1, NE/K004956/1 and NE/K004956/2. The use of the SRH1/MARFA radar data was funded by the Weston Family Foundation and G. Unger Vetlesen Foundation. Further, we acknowledge the use of the CReSIS MCoRDS3 data generated with support from the University of Kansas, NASA Operation IceBridge grant NNX16AH54G, NSF grants ACI-1443054, OPP-1739003, and IIS-1838230, Lilly Endowment Incorporated, and Indiana METACyt Initiative. We thank Kang Yang and Yao Lu for providing the DIC supraglacial rivers dataset. We also thank the Editor, Olaf Eisen, and the two anonymous reviewers for their constructive comments that helped improve the manuscript. K.C. was supported by the NASA Texas Space Grant Consortium Fellowship and the G. Unger Vetlesen Foundation. A.R. was supported by the UTIG postdoctoral fellowship program, the G. Unger Vetlesen Foundation, and the Greenland Climate Network (GC-Net) monitoring program. This is UTIG contribution number 3954.

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
