# Peer review of "Spatial characterization of near-surface structure and meltwater runoff conditions across Devon Ice Cap from dual-frequency radar reflectivity"

_The Cryosphere, 2022_

## Author Comment (AC1)

**Response to RC1: Spatial characterization of near-surface structure and meltwater runoff conditions across Devon Ice Cap from dual-frequency radar reflectivity**

Dear Anonymous Reviewer,

We thank you very much for your review of the manuscript. The comments/suggestions (italicized below) were very helpful and constructive for improving this work. We've addressed each point below with our responses and proposed revisions provided as bulleted text.

Sincerely,
Kristian Chan, on behalf of the co-authors

*General Comments*

*Chan et al. investigate the surface coherent return power of reflected radar waves by applying the Radar Statistical Reconnaissance method to multiple ice penetrating radar datasets with different center frequencies over Devon Ice Cap (Canada). The data is used to better characterize the composition of the firn pack (and if the firn pack contains ice layers) of the upper meters over the ice cap, which is important to better understand melting and refreezing processes as well as meltwater infiltration and runoff. The measured reflectivities are compared with modeled reflectivities using a reflectivity model informed by existing information on the firn pack (from ground-based ice penetrating radar data and firn cores). Their results suggest meter-thick ice slabs in certain parts of the ice cap, which permits surface water runoff away from the ice cap.*

*Overall I find the study by Chan et al. to be informative and very well written. They applied a smart approach to characterize the firnpack with existing multiple airborne radar data sets and other auxiliary data sets (such as firn cores and land-based radar data). Although the methodology is not fundamentally new and many aspects have been already analyzed and built upon previous studies (such as in Rutishauser et al., 2016), I believe that this article deserves to be published in The Cryosphere.*

*The basis for my decision is that, in my opinion, this is a robust study that is well structured, clearly written, and represents a significant step forward in knowledge on which future studies can build on and which is very useful for the cryosphere community. What I particularly liked is that the authors use existing data sets and put the data into a new context with their method to find out more about the first meters of firn of the Devon Ice Cap. Below I have some comments and questions that I think might help to add clarity and make the article better readable and easier to follow.*

*Main Remarks*

*Introduction:*

*The introduction could benefit from a small introduction on the Devon ice cap and why it is a*

*particularly good place to characterize the firn column. Either in a new paragraph (which I would prefer) or incorporated in one of the existing paragraphs.*

- Thank you for the suggestion. We will add some introductory text on Devon Ice Cap and how percolation/refreezing has made it a good place for characterizing firn heterogeneity.

*Figure 1:*

*(1) It would be a nice addition to have an overview map of the Canadian arctic or Canadian-Greenlandic arctic pointing out the location of the survey area. This would give the reader a much better impression of where the Devon ice cap is located.*

- We will add an overview map that indicates the location of the Devon Ice Cap.

*(2) I also would suggest finding a better solution with the contour lines and their elevation labels. They appear very chaotic at the ice caps margins, which is rather confusing than helpful information. The same applies to all other figures (also in the supplement; S4) in which the contour lines are shown. Maybe only displaying contour lines only above 600 m would make the plot less overloaded.*

- Thank you for the suggestion. We will clean up the contour lines at the ice cap margins by including contour lines above 600 m elevation (or a similar solution).

*(3) It would also be good to state what kind of satellite image you are using as a background image.*

- This was a Landsat image. We will include this information in the caption as well.

*Table 1:*

*Please explain the symbols in the table caption (e.g., that range resolution is z_0, etc.)*

*In addition, but very minor: a hline between the two systems would be nice to immediately see which z0 belongs to which system.*

- We will redefine the symbols in the caption and add the hline in the table.

*Figure 2:*

*What about the following idea: To give the reader a better understanding of the different depth resolution of the radar systems and which parts of the firn column are affected, one idea would be to somehow draw or indicate the depths that HiCARS & MARFA and MCoRDS3 resolve in Figure 2b.*

- Thank you for this suggestion. Resolution depths for HiCARS were initially included but removed from the plot, because they relied on assumptions made about the firn column,

such as firn permittivity/density and layer thickness. This was a motivation for including Table 1 in the main text. However, we agree that having this drawn on the figure could be very useful for visualizing the resolution depth. One option would be to add a resolution depth and indicate somewhere, either in the caption or main text, the assumptions used to calculate that depth. We can include this and/or explore options for how to best communicate the depth resolutions of each radar system while trying to make the overall figure as clear as possible to the reader.

*Figure 3:*

*(1) I think the figure could be better arranged if, for example, (a) and (b) were in a row and (c) below. Then the subfigures would be bigger and the whole figure would take probably less space in the document at the same time. The same could be done with Figure 4.*

- Agreed, we will rearrange the subplots in Figures 3 and 4 as suggested.

*(2) Shouldn't the label of the colorbar be "dB" instead of "db"?*

- Yes, we will make this correction.

*(3) I would suggest a different color scale, preferably linear rather than divergent. This is because in the HiCARS display, for example, the transition from -10 to -15 dB is shown as a weak color change, while from -20 to -25 dB there is a strong color change (yellow to blue). Therefore, I would suggest a linear graded color scale to better interpret the changes in dB based on a color scale across the different data sets.*

- We chose the current color scale, because it is a colorblind-friendly option and broadly captures the changes of Pc across all the surveys and their relation to the zone boundaries. However, we will try to use a linear scale if it can better represent these data while maintaining consistency across the surveys.

*Figure 4:*

*Caption: define again that interquartile ranges is IQR and P_c surface coherent power (as in Fig. 3).*

- We will redefine these in the caption.

Discussion:

*I have a question regarding the ice slab thicknesses in Zone II. In Line 336 you state that the HiCARS/MARFA system captures the entire thickness of the ice slabs. Maybe I have missed it, but why is that the case and how do you know that the ice slabs along these radar profiles are not thicker than the range resolution of the system?*

*My next question is very similar and refers to the average ice slab thicknesses. You calculated a*

*mean ice slab thickness based on the range resolution of the two different radar (groups). Wouldn't it be rather a minimum average ice slab thickness? Because since you are only analyzing the surface return within the limits of the range resolution of the radar system, you cannot estimate if the ice slab continues with depth and is thicker, right?*

*For me it seems that based on the surface GPR data it is assumed that the ice slabs in this region are not thicker as what is for example shown in Figure 2b. However, it might nevertheless be possible that thicker ice slabs might be present along the airborne radar profiles where no surface radar data exists. I think this should be clarified and also mentioned in the uncertainty section.*

- Thank you for the comments and suggestions. On Devon Ice Cap, the Zone II/III boundary represents the transition from a region with firn to one without firn, which is also validated by the Landsat imagery (Fig. 5). This spatial boundary also represents where the maximum ice slab thickness is obtained over Devon Ice Cap, because ice slabs grow in thickness from higher to lower elevations (e.g., MacFerrin et al., 2019) but shouldn't exist beyond the Zone II/III boundary due to the lack of firn. In other words, the Zone II/III boundary constrains the maximum ice slab thickness on Devon Ice Cap. In addition, we believe that our derived ice slab thickness represents an average range of values for Devon Ice Cap. If ice slabs are thicker than the range resolution of HiCARS/MARFA in Zone II, we would expect a change in the pattern of Pc, particularly near the Zone II/III boundary. However, it remains fairly consistent throughout and thus also consistent with the interpretation that HiCARS/MARFA observes a 4-layer medium in Zone II. However, we do acknowledge in other regions, particularly in Greenland, ice slabs can certainly be thicker than the range resolution of both radar systems. In that case, the average ice slab thickness derived via this method could represent a minimum average, depending on the location of the firn line and how Pc behaves near this boundary. We will add some text to discuss the uncertainties/limitations of the approach, as suggested.

  MacFerrin, M., Machguth, H., As, D. V., Charalampidis, C., Stevens, C. M., Heilig, A., ... & Abdalati, W. (2019). Rapid expansion of Greenland's low-permeability ice slabs. *Nature, 573*(7774), 403-407.

*Figure 5:*

*Here now appears a reference to the background satellite image, but the coordinates are missing. Again, I would prefer to get rid of the contour lines and labels below a certain depth.*

- We will include coordinates and clean up the contour lines as previously mentioned.

*Supplement*

*Figure S4: Please mention once more in the caption that P_c is coherent specular and P_n incoherent/scattered. I'm sure many readers don't, but I often have the problem that I forget the*

*abbreviations in the text while reading and then have to look for them again in the text when they appear in a figure.*

- Agreed, especially since this is a supplemental figure. We will define Pc and Pn in the caption.

*Line-item Comments*

*L 84-86: I think that Operation Ice Bridge should be mentioned here as well in addition to the University of Kansas. Moreover, I would suggest using the acronym MCoRDS3 instead of just MCoRDS throughout the document.*

- We will mention Operation Ice Bridge here and change MCoRDS to MCoRDS3 throughout the manuscript.

*L 99-101: With respect to the factors affecting permittivity, I think that temperature and the anisotropy due to the orientation of the ice crystal fabric should also be mentioned (although COF may not be so important in the firn column). In that sense you could additionally cite for example Fujita et al. (2000):*

*"Fujita, S.,T. Matsuoka,T. Ishida,K. Matsuoka, and S. Mae (2000), A summary of the complex dielectric permittivity of ice in the megahertz range and its applications for radar sounding of polar ice sheets, in Physics of Ice Core Records, edited by T. Hondoh, pp. 185–212, Hokkaido Univ. Press, Hokkaido, Japan. "*

- Agreed, we will include temperature and ice crystal fabric as factors affecting permittivity, citing Fujita et al., 2000.

*L 128 (and L177-178): You mention that "[...] surface roughness is not the main contributor to surface scattering over DIC (Rutishauser et al., 2016).". It would be interesting to mention in one sentence why this is not the case. Especially because this assumption is important for the interpretation of the results.*

- Thank you for this comment. We agree that this is an important assumption for interpreting the results. Rutishauser et al., 2016 showed that the incoherent power is mainly governed by volume scattering from the ice layers as opposed to surface roughness.

  Looking at Figure 3 of Rutishauser et al., 2016, the majority of the laser-derived roughness values are concentrated at $\sigma_h = 0.05$ m. Propagating this value into Eq. 2 of this manuscript, specifically the exponential part of the equation representing the effects of surface roughness, we find that this contributes 0.07 dB to the coherent power (Pc). On the other hand, the effects of $r^2$ vary on the order of tens of dB (Figures 3 and S5). Moreover, the $\sigma_h$ values from laser altimetry in Rutishauser et al., 2016 were derived using a baseline of 171.5 m. We expect the surface roughness at the wavelength-scale of interest ($\lambda = 5$ m) to be much smaller, because the surface roughness scales with the

baseline length scale of interest (Shepard et al., 2001). Thus, the 0.07 dB surface roughness contribution to Pc already represents a highly conservative value. We can incorporate this calculation into the manuscript as well.

*L 137-139: Here you state that: "Previous applications of the RSR method have empirically shown that an aircraft roll of 2 to 3° allows for a stable coherent radar return." Is there a reference for this?*

- There is no reference for this at the moment. However, we will include this analysis and make it available, either here or elsewhere with a reference.

*L 141: The airborne radar data in your study is also "ground-penetrating". From what I understood you refer to land-based or surface radar in this section. Therefore I would suggest making clear that all radar surveys are ground penetrating and some are airborne and this one is land-based/surface radar data.*

- Thank you for pointing this out. We will clarify the terminology here to distinguish between land-based/surface radar and airborne ice-penetrating radar.

*L 248-252: I am not sure if I missed it, but is the difference between the old and refined Zones shown somewhere? If not, I think it should be (maybe in the Supplement). I guess the old Zones are those displayed in Rutishauser et al. (2016) in Figures 1a and 2?*

- That is correct. The old zone boundaries are those in Figure 2 of Rutishauser et al., 2016 but currently not shown in this work. We will include the old and new zone boundaries in the Supplement section.

*L 252-254: Here you refer to the Discussion Section but I think it would be also good to refer to Figure 5.*

- We will update this to point to Figure 5.

---

## Author Comment (AC2)

**Response to RC2: Spatial characterization of near-surface structure and meltwater runoff conditions across Devon Ice Cap from dual-frequency radar reflectivity**

Dear Anonymous Reviewer,

We thank you very much for your review of the manuscript. The comments/suggestions (italicized below) were very helpful and constructive for improving this work. We've addressed each point below with our responses and proposed revisions provided as bulleted text.

Sincerely,
Kristian Chan, on behalf of the co-authors

*This study uses four radar datasets (3 airborne, 1 ground based) to evaluate the firn characteristics of Devon Ice Cap in the Canadian Arctic. The general characteristics of the firn were already classified using the ground-based dataset, and the new element here is using all the airborne data together to look at the firn. These data are used as a way to assess the spatial distribution of firn properties in more detail, within the general framework of the ground-based survey. Conclusions about ice-slab thickness and melt channel distribution are derived largely using the variability in return power of the surveys within different "zones" of firn, relying on the ground-based survey to get the general structure (i.e. large slabs, thin lenses, etc). The implications for meltwater runoff are discussed, making a nice story. The main novel element here is inferring properties of ice lenses in firn using multiple airborne radars that do not resolve the ice lenses/slabs explicitly, but instead have some return-power sensitivity to the near-surface properties.*
*This study is novel, generally well written, and well-suited to The Cryosphere. I have two major comments and a variety of small points that I think are important to address before publication, but then I think it should be a nice contribution.*

*Major Comments:*

*There is insufficient analysis of whether one could conduct a similar study in the absence of some independent radar measurements that actually resolve the bottom of the ice slabs (i.e. the GPR)—perhaps this was never the goal of the study, but the title and some of the language suggest otherwise, which I think sets the reader up to be dissatisfied at what is otherwise a nice paper. The suggestion in the title, abstract, and conclusions is that the dual-frequency reflectometry can be used on its own to garner insight into firn properties (and extra-terrestrial applications cannot rely on such validation). As I read the paper, the analysis of things like the ice-slab thickness in Zone II (Section 3.2.3 and Discussion) relies on already knowing that this area has thick ice slabs, and otherwise the variations could be misinterpreted as density variations or similar. If the paper can be altered to use the GPR as validation rather than as a necessary component, that would be ideal; for example, is there some objective measure that would allow the picking of the zone boundaries from these model results? I assume that the answer is no since otherwise it would be discussed (which is worth adding to the text); I think this study will merit publication without that analysis, although in this case I think textual/title alterations are needed throughout to make clear that what is really happening is analysis of things like ice-slab thickness when the general firn structure (zonal classification in this case)*

*already independently known, effectively requiring a third radar dataset (GPR) or other extensive in-situ measurements.*

*I find Section 3.1 to be lacking in purpose, in part because it reads something like a failed attempt to distinguish the zonal classification based solely on reflectometry; it is doubly unconvincing due to insufficient error analysis. In lines 201-203 there are claims about which model fits better where, but there is not even an analysis of the relative RMS misfits of the two models in the two zones. At a bare minimum, such basic model-data misfit analysis is needed to make any claim about what model fits where. However, given the section title I was hoping it would essentially answer the other main point raised above. I understand that this may be beyond the scope of the work or not supported by it, but then I am left wondering what this section really adds (perhaps adding some error analysis would change my mind, and I could better understand what we could conclude out of this section). Perhaps some roadmap under the general "Results" heading could help as well.*

- Thank you for the comments and suggestions. We agree that there are limitations to this method, particularly without GPR measurements. However, with dual-frequency reflectometry on its own, one would be able to determine if layering is present in the near-surface firn, because in the case with layering, the radar response is dispersive (i.e., frequency-dependent). For example, if we consider the case of homogenous firn without layers, the assumption made is that the coherent power (Pc) is mainly sensitive to surface density variations. In this case, the radar response is non-dispersive, because the strongest reflection is that from the surface, and mono-frequency radar data is sufficient to invert the surface return for density. However, a dual-frequency system would be able to confirm whether Pc is mainly affected by surface density or the presence of ice slabs, because Pc would appear to be the same for both radar returns in the absence of ice slabs. For this work, one of the goals was to apply this dual-frequency approach to show that, indeed, the coherent power is not representative of surface density. In regions without a priori knowledge of the general firn structure, the dual-frequency method would provide insight into the presence of ice slabs at characteristic depths within the near-surface (if both systems utilize different bandwidths).

  The firn zone boundaries were derived completely independent of the GPR measurements, by comparing the balance between the coherent and incoherent power of the total surface power recorded by the MARFA airborne radar. What we find are changes of the near-surface structure consistent with these zonal boundaries, validated by the GPR data and imagery as well. To better communicate that this auxiliary GPR data was used for validation, we will reorganize section 3 by moving current Section 3.1 to the last part of this section. We will also rename current Section 3.1 to reflect its purpose in this study, which is to serve as ground-truth and validation of our interpretation of the dual-frequency airborne radar datasets. Thus, the dual-frequency airborne radar results would then be the focus of this section. The thin layer model was developed also for validation purposes and does not form the main focus of this section, although we do provide some error/sensitivity analysis of ice slab thickness (from the GPR) and firn density (from the firn cores) as inputs to the model. We believe that this is sufficient for the purposes of the model and additional error analysis is beyond the scope of this work.

To better highlight what we can learn in the absence of measurements such as GPR, we will add additional text to discuss the advantages of a dual-frequency system compared to a mono-frequency system, as mentioned above, and a roadmap in the 'Results' section as suggested. We will also clarify the limitations of this method for future applications to other regions of interest. We believe that these edits would hopefully make clear the purpose of these subsections and the overall goals of this work.

*Line Comments:*

*52: I would suggest removing the IPR acronym. These are all ice-penetrating radars, and the terminology is unnecessarily confusing.*

- Thank you for the suggestion. We will make clear whether we are referring to land/surface-based vs. airborne ice-penetrating radar in the text.

*57: What such methods? The low frequency ones?*

- Yes, this is referring to low frequency methods for near-surface characterization. We will clarify this in the text.

*58: I am skeptical of this claim—does Mars have surface melt? Could ice lenses and slabs be possible? While other dual-frequency applications matter there, the relevance of this study should be justified or the line should be deleted.*

- Here, we are only referring to general near-surface properties on Mars (e.g., thin surficial layering of $CO_2$ ice) investigated with lower radar frequencies. The main idea here is to refer to studies where near-surface properties can be studied even if features cannot be directly resolved.

*69: What is compact ice? It is not defined nor is it a common term. I think it just means glacier ice as opposed to firn. Perhaps "fully compacted" or "fully densified" would be more appropriate. While I put this as a line comment, I think it is important to change "compact" throughout the paper, since it is not quite the technical term and the word has multiple plain-language meanings.*

- Thank you for pointing this out. We will change the terminology here and throughout the manuscript.

*82: What does dual phase mean?*

- It refers to the ability to record data from either the left or right antennas separately on the survey plane (see Scanlan et al., 2020 and Young et al., 2016), which is the main difference between the HiCARS and MARFA systems. However, this does not change the interpretation and analysis of the data for the purposes of this work.

Scanlan, K. M., Rutishauser, A., Young, D. A., & Blankenship, D. D. (2020). Interferometric discrimination of cross-track bed clutter in ice-penetrating radar sounding data. Annals of Glaciology, 61(81), 68-73.

Young, D. A., Schroeder, D. M., Blankenship, D. D., Kempf, S. D., & Quartini, E. (2016). The distribution of basal water between Antarctic subglacial lakes from radar sounding. Philosophical Transactions of the Royal Society A: Mathematical, Physical and Engineering Sciences, 374(2059), 20140297.

*99: There are plenty of homogeneous media for which the arguments in line 100 apply—perhaps just delete this sentence*

- We will remove this sentence.

*Table 1: The layout here is confusing. I think I would have understood better if the epsilon_eff column were deleted and there were separate columns for $z_0$ for firn and for ice. Also should specify that this is not a universal firn number—it assumes 410 kg/m$^3$ or something similar.*

- Thank you for the suggestions. We will reformat this table to make it easier for the reader and specify the corresponding density value that is representative of the chosen permittivity for firn in this particular case.

*115: It would be helpful to have a half sentence about why the bin size (in spatial terms) is different for the different systems.*

- We will include this information.

*126: RMS height of what? I guess this should be surface elevation*

- It is the standard deviation of the surface topography measured along a profile (see Shepard et al., 2001).

  Shepard, M. K., Campbell, B. A., Bulmer, M. H., Farr, T. G., Gaddis, L. R., & Plaut, J. J. (2001). The roughness of natural terrain: A planetary and remote sensing perspective. Journal of Geophysical Research: Planets, 106(E12), 32777-32795.

*129: The hypothesis that the return power variation is dominated by variations in $r^2$ is a large and critical assumption that is brushed aside too flippantly. I guess there was some work in Rutishauser et al., 2016, to justify that it is not dominant, but I think it is a bit too important to be relegated to a reference, since strong dependence on the roughness may invalidate any conclusions. Addressing this could be as simple as estimating the maximum variation resulting from a realistic range of roughnesses compared to the variation in return power.*

- Thank you for the comment. We agree that this is an important assumption, as strong dependence on surface roughness will affect the coherency of the signal. Looking at Figure 3 of Rutishauser et al., 2016, the majority of the laser-derived roughness values

are concentrated at $\sigma_h = 0.05$ m. Propagating this value into Eq. 2 of this manuscript, specifically the exponential part of the equation representing the effects of surface roughness, we find that this contributes 0.07 dB to the coherent power (Pc). On the other hand, the effects of $r^2$ vary on the order of tens of dB (Figures 3 and S5). Moreover, the $\sigma_h$ values from laser altimetry in Rutishauser et al., 2016 were derived using a baseline of 171.5 m. We expect the surface roughness at the wavelength-scale of interest ($\lambda = 5$ m) to be much smaller, because the surface roughness scales with the baseline length scale of interest (Shepard et al., 2001). Thus, the 0.07 dB surface roughness contribution to Pc already represents a highly conservative value. We can incorporate this calculation into the manuscript as well.

*139: Is this a typo? Why exclude rock based on aircraft elevation rather than imagery, etc.*

- This is ambiguous. We will clarify that this refers to surface elevation and not aircraft elevation.

*143: At least a brief overview of the GPR system belongs here—the reader should not have to go to Rutishauser et al. just to find out the frequency*

- We will add some relevant details of the GPR system here.

*155: Layers of what, and should this be i.e.? Generally I would assume density is the only important factor in such shallow reflections—if not, what else should be included.*

- Thank you for the question. This refers to any type of layering within the near-surface that results in a dielectric contrast strong enough to generate interference with the return from the surface. This can be governed by density variations but also any phase changes in the firn, such meltwater/brine. We will remove the parentheses and its text, because this sentence should be agnostic to any specific assumptions of the near-surface in this part of the section.

*326: potentially insightful*

- We will make this edit in the text.

*391: Rephrase slightly to clarify that the ambiguity is due to tradeoffs between density and layer thickness*

- We will reword this line to mention this.

*395: Caution against?*

- We will make this edit in the text.

*411-420: I would highly recommend moving this paragraph upward into discussion—I do not find it to be particularly convincing, and I don't really think it is a conclusion as such. It is not*

*my place to demand such a change, but take this as a stylistic suggestion of a way to make the paper more impactful.*

- Thank you for the suggestion. We will explore where in the discussion section this paragraph might fit better.

---

## Author Response (AR1)

**Spatial characterization of near-surface structure and meltwater runoff conditions across Devon Ice Cap from dual-frequency radar reflectivity**
Chan et al.

*Locations where 'revised' is mentioned below, where applicable, refer to locations in the revised clean manuscript (i.e., without tracked changes).*

Response to comments from Reviewer 1:

| | |
|---|---|
| Comment | The introduction could benefit from a small introduction on the Devon ice cap and why it is a particularly good place to characterize the firn column. Either in a new paragraph (which I would prefer) or incorporated in one of the existing paragraphs. |
| Location | Introduction |
| Response | We added some introductory text on Devon Ice Cap in a new paragraph. |

| | |
|---|---|
| Comment | It would be a nice addition to have an overview map of the Canadian arctic or Canadian-Greenlandic arctic pointing out the location of the survey area. This would give the reader a much better impression of where the Devon ice cap is located. |
| Location | Figure 1 |
| Response | We added an overview map to Figure 1 that indicates the location of Devon Ice Cap. |

| | |
|---|---|
| Comment | I also would suggest finding a better solution with the contour lines and their elevation labels. They appear very chaotic at the ice caps margins, which is rather confusing than helpful information. The same applies to all other figures (also in the supplement; S4) in which the contour lines are shown. Maybe only displaying contour lines only above 600 m would make the plot less overloaded. |
| Location | Figure 1 |
| Response | We cleaned up the contour lines at the ice cap margins by only including contour lines above 600 m elevation for all figures where applicable. |

| | |
|---|---|
| Comment | It would also be good to state what kind of satellite image you are using as a background image. |
| Location | Figure 1 |
| Response | This was a Landsat image. We included this information in the caption. |

| Comment | Please explain the symbols in the table caption (e.g., that range resolution is z_0, etc.). In addition, but very minor: a hline between the two systems would be nice to immediately see which z0 belongs to which system. |
|---|---|
| Location | Table 1 |
| Response | We redefined the symbols in the caption and added the hline in the table. |

| Comment | What about the following idea: To give the reader a better understanding of the different depth resolution of the radar systems and which parts of the firn column are affected, one idea would be to somehow draw or indicate the depths that HiCARS & MARFA and MCoRDS3 resolve in Figure 2b. |
|---|---|
| Location | Figure 2 (initial) / Figure 4 (revised) |
| Response | Resolution depths for HiCARS2 were initially included but removed from the plot, because they relied on assumptions made about the firn column, such as firn permittivity/density and layer thickness. This was a motivation for including Table 1 in the main text. However, we agree that having this drawn on the figure could be very useful for visualizing the resolution depth. We added resolution depths for HiCARS2/MARFA and MCoRDS3, and indicated in the caption the assumptions used to calculate that depth. |

| Comment | I think the figure could be better arranged if, for example, (a) and (b) were in a row and (c) below. Then the subfigures would be bigger and the whole figure would take probably less space in the document at the same time. The same could be done with Figure 4. |
|---|---|
| Location | Figures 3 and 4 (initial) / Figure 2 and 3 (revised) |
| Response | We rearranged the subplots for these two figures as suggested. |

| Comment | Shouldn't the label of the colorbar be "dB" instead of "db"? |
|---|---|
| Location | Figure 3 (initial) / Figure 2 (revised) |
| Response | We changed "db" to "dB". |

| | |
|---|---|
| Comment | I would suggest a different color scale, preferably linear rather than divergent. This is because in the HiCARS display, for example, the transition from -10 to -15 dB is shown as a weak color change, while from -20 to -25 dB there is a strong color change (yellow to blue). Therefore, I would suggest a linear graded color scale to better interpret the changes in dB based on a color scale across the different data sets. |
| Location | Figure 3 (initial) / Figure 2 (revised) |
| Response | We changed the color scale to a linear graded scale across the surveys. |

| | |
|---|---|
| Comment | Caption: define again that interquartile ranges is IQR and P_c surface coherent power (as in Fig. 3). |
| Location | Figure 4 (initial) / Figure 3 (revised) |
| Response | We redefined these in the caption. |

| | |
|---|---|
| Comment | I have a question regarding the ice slab thicknesses in Zone II. In Line 336 you state that the HiCARS/MARFA system captures the entire thickness of the ice slabs. Maybe I have missed it, but why is that the case and how do you know that the ice slabs along these radar profiles are not thicker than the range resolution of the system?

My next question is very similar and refers to the average ice slab thicknesses. You calculated a mean ice slab thickness based on the range resolution of the two different radar (groups). Wouldn't it be rather a minimum average ice slab thickness? Because since you are only analyzing the surface return within the limits of the range resolution of the radar system, you cannot estimate if the ice slab continues with depth and is thicker, right?

For me it seems that based on the surface GPR data it is assumed that the ice slabs in this region are not thicker as what is for example shown in Figure 2b. However, it might nevertheless be possible that thicker ice slabs might be present along the airborne radar profiles where no surface radar data exists. I think this should be clarified and also mentioned in the uncertainty section. |
| Location | Discussion |
| Response | On Devon Ice Cap, the Zone II/III boundary represents the transition from a region with firn to one without firn, which is validated by the Landsat imagery (Fig. 5). This spatial boundary also represents where the maximum ice slab thickness is obtained over Devon Ice Cap, because ice slabs grow in thickness from higher to lower elevations (e.g., MacFerrin et al., 2019) but shouldn't exist beyond the Zone II/III boundary due to the lack of firn. In other words, the Zone II/III boundary constrains the maximum ice slab thickness on Devon Ice Cap. In addition, we believe that our derived ice slab thickness represents an average range of values for Devon Ice Cap. If ice slabs are thicker than the range resolution of HiCARS2/MARFA in Zone II, we would expect a change in the pattern of Pc, particularly near the Zone II/III boundary. However, it remains fairly consistent throughout and thus also consistent with the interpretation that HiCARS2/MARFA observes a 4-layer medium in Zone II. However, we do acknowledge |

| | |
|---|---|
| | in other regions, particularly in Greenland, ice slabs can certainly be thicker than the range resolution of both radar systems. In that case, the average ice slab thickness derived via this method could represent a minimum average, depending on the location of the firn line and how Pc behaves near this boundary. We added some text to discuss the uncertainties/limitations of the approach, as suggested, in a new Section 4.2 within the Discussion section. |

| | |
|---|---|
| Comment | Here now appears a reference to the background satellite image, but the coordinates are missing. Again, I would prefer to get rid of the contour lines and labels below a certain depth. |
| Location | Figure 5 |
| Response | We added coordinates and removed contour lines below 600 m elevation. |

| | |
|---|---|
| Comment | Figure S4: Please mention once more in the caption that P_c is coherent specular and P_n incoherent/scattered. I'm sure many readers don't, but I often have the problem that I forget the abbreviations in the text while reading and then have to look for them again in the text when they appear in a figure. |
| Location | Figure S4 (initial) / Figure S2 (revised) |
| Response | We defined Pc and Pn in the caption. |

| | |
|---|---|
| Comment | I think that Operation Ice Bridge should be mentioned here as well in addition to the University of Kansas. Moreover, I would suggest using the acronym MCoRDS3 instead of just MCoRDS throughout the document. |
| Location | Line 84-86 (initial) / Line 94-96 (revised) |
| Response | We added Operation Ice Bridge and changed MCoRDS to MCoRDS3 throughout the manuscript. We also changed HiCARS to HiCARS2. |

| | |
|---|---|
| Comment | With respect to the factors affecting permittivity, I think that temperature and the anisotropy due to the orientation of the ice crystal fabric should also be mentioned (although COF may not be so important in the firn column). In that sense you could additionally cite for example Fujita et al. (2000) |
| Location | Line 99-100 (initial) / Line 110-111 (revised) |
| Response | We included temperature and ice crystal fabric as factors affecting permittivity, citing Fujita et al., 2000. |

| | |
|---|---|
| Comment | You mention that "[...] surface roughness is not the main contributor to surface scattering over DIC (Rutishauser et al., 2016).". It would be interesting to mention in one sentence why this is not the case. Especially because this assumption is important for the interpretation of the results. |
| Location | Line 128, 177-178 (initial) / Line 137-146 (revised) |
| Response | We agree that this is an important assumption for interpreting the results. Rutishauser et al., 2016 showed that the incoherent power is mainly governed by volume scattering from the ice layers as opposed to surface roughness. Looking at Figure 3 of Rutishauser et al., 2016, the laser-derived roughness values in Zone II are concentrated at $\sigma_h$ = 0.09 m. Propagating this value into Eq. 2 of this manuscript, specifically the exponential part of the equation representing the effects of surface roughness, we find that this contributes 0.22 dB to the coherent power (Pc). On the other hand, the effects of $r^2$ vary on the order of tens of dB (Figures 2 and S3). Moreover, the $\sigma_h$ values from laser altimetry in Rutishauser et al., 2016 were derived using a baseline of 171.5 m. We expect the surface roughness at the wavelength-scale of interest ($\lambda$ = 5 m) to be much smaller, because the surface roughness scales with the baseline length scale of interest (Shepard et al., 2001). Thus, the 0.22 dB surface roughness contribution to Pc already represents a highly conservative value. We added this calculation and explanation into the manuscript. |

| | |
|---|---|
| Comment | Here you state that: "Previous applications of the RSR method have empirically shown that an aircraft roll of 2 to 3° allows for a stable coherent radar return." Is there a reference for this? |
| Location | Line 137-139 (initial) / Line 158 (revised) |
| Response | We included the justification for this and its reference. |

| | |
|---|---|
| Comment | The airborne radar data in your study is also "ground-penetrating". From what I understood you refer to land-based or surface radar in this section. Therefore I would suggest making clear that all radar surveys are ground penetrating and some are airborne and this one is land-based/surface radar data. |
| Location | Line 141 (initial) / Line 161 (revised) |
| Response | We clarified the terminology to distinguish between surface-based radar and airborne ice-penetrating radar through the manuscript. |

| Comment | I am not sure if I missed it, but is the difference between the old and refined Zones shown somewhere? If not, I think it should be (maybe in the Supplement). I guess the old Zones are those displayed in Rutishauser et al. (2016) in Figures 1a and 2? |
|---|---|
| Location | Line 248-252 (initial) / Line 234-238 (revised) |
| Response | That is correct. The old zone boundaries are those in Figure 2 of Rutishauser et al., 2016. We included the old and new zone boundaries in the Supplement section, Figure S2. |

| Comment | Here you refer to the Discussion Section but I think it would be also good to refer to Figure 5. |
|---|---|
| Location | Line 252-254 (initial) / Line 238-240 (revised) |
| Response | We updated this to point to Figure 5. |

**Response to comments from Reviewer 2:**

| | |
|---|---|
| Comment | There is insufficient analysis of whether one could conduct a similar study in the absence of some independent radar measurements that actually resolve the bottom of the ice slabs (i.e. the GPR)—perhaps this was never the goal of the study, but the title and some of the language suggest otherwise, which I think sets the reader up to be dissatisfied at what is otherwise a nice paper. The suggestion in the title, abstract, and conclusions is that the dual-frequency reflectometry can be used on its own to garner insight into firn properties (and extra-terrestrial applications cannot rely on such validation). As I read the paper, the analysis of things like the ice-slab thickness in Zone II (Section 3.2.3 and Discussion) relies on already knowing that this area has thick ice slabs, and otherwise the variations could be misinterpreted as density variations or similar. If the paper can be altered to use the GPR as validation rather than as a necessary component, that would be ideal; for example, is there some objective measure that would allow the picking of the zone boundaries from these model results? I assume that the answer is no since otherwise it would be discussed (which is worth adding to the text); I think this study will merit publication without that analysis, although in this case I think textual/title alterations are needed throughout to make clear that what is really happening is analysis of things like ice-slab thickness when the general firn structure (zonal classification in this case) already independently known, effectively requiring a third radar dataset (GPR) or other extensive in-situ measurements. |
| | I find Section 3.1 to be lacking in purpose, in part because it reads something like a failed attempt to distinguish the zonal classification based solely on reflectometry; it is doubly unconvincing due to insufficient error analysis. In lines 201-203 there are claims about which model fits better where, but there is not even an analysis of the relative RMS misfits of the two models in the two zones. At a bare minimum, such basic model-data misfit analysis is needed to make any claim about what model fits where. However, given the section title I was hoping it would essentially answer the other main point raised above. I understand that this may be beyond the scope of the work or not supported by it, but then I am left wondering what this section really adds (perhaps adding some error analysis would change my mind, and I could better understand what we could conclude |

| | |
|---|---|
| | out of this section). Perhaps some roadmap under the general "Results" heading could help as well. |
| Location | General |
| Response | We agree that there are limitations to this method, particularly without GPR measurements. However, with dual-frequency reflectometry on its own, one would be able to determine if layering is present in the near-surface firn, because in the case with layering, the radar response is dispersive (i.e., frequency-dependent). For example, if we consider the case of homogenous firn without layers, the assumption made is that the coherent power (Pc) is mainly sensitive to surface density variations. In this case, the radar response is non-dispersive, because the strongest reflection is that from the surface, and mono-frequency radar data is sufficient to invert the surface return for density. However, a dual-frequency system would be able to confirm whether Pc is mainly affected by surface density or the presence of ice slabs, because Pc would appear to be the same for both radar returns in the absence of ice slabs. For this work, one of the goals was to apply this dual-frequency approach to show that, indeed, the coherent power is not representative of surface density. In regions without a priori knowledge of the general firn structure, the dual-frequency method would provide insight into the presence of ice slabs at characteristic depths within the near-surface (if both systems utilize different bandwidths).

The firn zone boundaries were derived completely independent of the GPR measurements, by comparing the balance between the coherent and incoherent power of the total surface power recorded by the MARFA airborne radar. What we find are changes of the near-surface structure consistent with these zonal boundaries, validated by the GPR data and imagery as well. To better communicate that this auxiliary GPR data was used for validation, we reorganized section 3 by moving Section 3.1 (of the initial manuscript) to the last part of the Results section. We also renamed initial Section 3.1 to reflect its purpose in this study, which is to serve as ground-truth and validation of our interpretation of the dual-frequency airborne radar datasets. Thus, the dual-frequency airborne radar results would then be the focus of the Results section. The thin layer model was developed also for validation purposes and does not form the main focus of this section, although we do provide some error/sensitivity analysis of ice slab thickness (from the GPR) and firn density (from the firn cores) as inputs to the model. We believe that this is sufficient for the purposes of the model and additional error analysis is beyond the scope of this work.

To better highlight what we can learn in the absence of measurements such as GPR, we added new Section 4.2, to discuss the advantages of a dual-frequency system compared to a mono-frequency system, as mentioned above, and a roadmap under the general Results heading as suggested. We also clarified the limitations of this method for future applications to other regions of interest in Section 4.2. We believe that these edits would hopefully make clear the purpose of the Results subsections and the overall goals of this work. |

| Comment | I would suggest removing the IPR acronym. These are all ice-penetrating radars, and the terminology is unnecessarily confusing. |
| --- | --- |
| Location | Line 52 (initial) |
| Response | We opted to keep the IPR acronym but clarified throughout the manuscript whether we are referring to surface-based radar or airborne ice-penetrating radar. We also removed the text in parentheses. |

| Comment | What such methods? The low frequency ones? |
| --- | --- |
| Location | Line 57 (initial) / Line 63 (revised) |
| Response | Yes, this is referring to low frequency methods for near-surface characterization. We clarified this in the text. |

| Comment | I am skeptical of this claim—does Mars have surface melt? Could ice lenses and slabs be possible? While other dual-frequency applications matter there, the relevance of this study should be justified or the line should be deleted. |
| --- | --- |
| Location | Line 58 (initial) / Line 62 (revised) |
| Response | Here, we are only referring to general near-surface properties on Mars (e.g., thin surficial layering of $CO_2$ ice) investigated with lower radar frequencies. The main idea here is to refer to studies where near-surface properties can be studied even if features cannot be directly resolved. We clarified this in the text. |

| Comment | What is compact ice? It is not defined nor is it a common term. I think it just means glacier ice as opposed to firn. Perhaps "fully compacted" or "fully densified" would be more appropriate. While I put this as a line comment, I think it is important to change "compact" throughout the paper, since it is not quite the technical term and the word has multiple plain-language meanings. |
| --- | --- |
| Location | Line 69 (initial) / Line 72 (revised) |
| Response | We changed the terminology here and throughout the manuscript. |

| Comment | What does dual phase mean? |
| --- | --- |
| Location | Line 82 (initial) / Line 92 (revised) |
| Response | It refers to the ability to record data from either the left or right antennas separately on the survey plane (see Scanlan et al., 2020 and Young et al., 2016), which is the main |

| | difference between the HiCARS2 and MARFA systems. However, this does not change the interpretation and analysis of the data for this work. |
|---|---|

| | |
|---|---|
| Comment | There are plenty of homogeneous media for which the arguments in line 100 apply—perhaps just delete this sentence. |
| Location | Line 100 (initial) |
| Response | We removed this sentence. |

| | |
|---|---|
| Comment | The layout here is confusing. I think I would have understood better if the epsilon_eff column were deleted and there were separate columns for z0 for firn and for ice. Also should specify that this is not a universal firn number—it assumes 410 kg/m3 or something similar. |
| Location | Table 1 |
| Response | We reformatted this table to make it easier for the reader and specified the corresponding density value that is representative of the chosen permittivity for firn in this particular case. |

| | |
|---|---|
| Comment | It would be helpful to have a half sentence about why the bin size (in spatial terms) is different for the different systems. |
| Location | Line 115 (initial) / Line 123-125 (revised) |
| Response | We added this information. |

| | |
|---|---|
| Comment | RMS height of what? I guess this should be surface elevation. |
| Location | Line 126 (initial) |
| Response | It is the standard deviation of the surface topography measured along a profile (see Shepard et al., 2001). |

| | |
|---|---|
| Comment | The hypothesis that the return power variation is dominated by variations in $r^2$ is a large and critical assumption that is brushed aside too flippantly. I guess there was some work in Rutishauser et al., 2016, to justify that it is not dominant, but I think it is a bit too important to be relegated to a reference, since strong dependence on the roughness may invalidate any conclusions. Addressing this could be as simple as estimating the maximum variation resulting from a realistic range of roughnesses compared to the variation in return power. |
| Location | Line 129 (initial) / Line 137-146 (revised) |
| Response | We agree that this is an important assumption, as strong dependence on surface roughness will affect the coherency of the signal. Looking at Figure 3 of Rutishauser et al., 2016, the the laser-derived roughness values in Zone II are concentrated at $\sigma_h = 0.09$ m. Propagating this value into Eq. 2 of this manuscript, specifically the exponential part of the equation representing the effects of surface roughness, we find that this contributes 0.22 dB to the coherent power (Pc). On the other hand, the effects of $r^2$ vary on the order of tens of dB (Figures 2 and S3). Moreover, the $\sigma_h$ values from laser altimetry in Rutishauser et al., 2016 were derived using a baseline of 171.5 m. We expect the surface roughness at the wavelength-scale of interest ($\lambda = 5$ m) to be much smaller, because the surface roughness scales with the baseline length scale of interest (Shepard et al., 2001). Thus, the 0.22 dB surface roughness contribution to Pc already represents a highly conservative value. We added this calculation and explanation into the manuscript. |

| | |
|---|---|
| Comment | Is this a typo? Why exclude rock based on aircraft elevation rather than imagery, etc. |
| Location | Line 139 (initial) / Line 159 (revised) |
| Response | We clarified that this refers to surface elevation and not aircraft elevation. |

| | |
|---|---|
| Comment | At least a brief overview of the GPR system belongs here—the reader should not have to go to Rutishauser et al. just to find out the frequency |
| Location | Line 143 (initial) / Line 163-164 (revised) |
| Response | We added some relevant details of the GPR system. |

| Comment | Layers of what, and should this be i.e.? Generally I would assume density is the only important factor in such shallow reflections—if not, what else should be included. |
|---|---|
| Location | Line 155 (initial) |
| Response | This refers to any type of layering within the near-surface that results in a dielectric contrast strong enough to generate interference with the return from the surface. This can be governed by density variations but also any phase changes in the firn, such as meltwater/brine. We removed the parentheses and text contained within, because this sentence should be agnostic to any specific assumptions of the near-surface in this part of the section. |

| Comment | "potentially insightful" |
|---|---|
| Location | Line 326 (initial) / Line 313 (revised) |
| Response | We made this edit in the text. |

| Comment | Rephrase slightly to clarify that the ambiguity is due to tradeoffs between density and layer thickness |
|---|---|
| Location | Line 391 (initial) / Line 426-427 (revised) |
| Response | We reworded this sentence and incorporated it into new Section 4.2. |

| Comment | Caution against? |
|---|---|
| Location | Line 395 (initial)/ Line 427 (revised) |
| Response | We changed this in the text. |

| Comment | I would highly recommend moving this paragraph upward into discussion—I do not find it to be particularly convincing, and I don't really think it is a conclusion as such. It is not my place to demand such a change, but take this as a stylistic suggestion of a way to make the paper more impactful. |
|---|---|
| Location | Line 411-420 (initial) / Line 444-450 (revised) |
| Response | We moved this to new Section 4.2 in the Discussion. |

---

## Editor Decision (ED1)

**Summary of Comments on Microsoft Word - Chan_Devon_TC_rev1.docx**

**Page: 1**

Author: oeisen    Subject: Hervorheben    Date: 11.03.23, 13:23:07
see general comment - delete.
airborne radar or airborne radio-echo sounding

Author: oeisen    Subject: Hervorheben    Date: 11.03.23, 13:25:38
higher
I don't get this argument in the abstract, because not mentioned before.
Undefined, what higher frequency means here, can't be HF. Do you mean UHF?
Or rather high-resolution (cm-dm scale)?
* * *
Author: oeisen     Subject: Hervorheben     Date: 11.03.23, 13:31:00

IPR is nor very often used. We recenlty argued to get rid of this term altogether (Schlegel et al., Ann. Glac., 2023). Ice-penetrating is nothing else than ground-penetrating, where the ground is made of ice. However, GPR is usually refered to as ground-based. This becomes obsolete now that GPRs are also flown underneath helicopters. I support the statement of the reviewer here.
To be more consistent with the most used convention I suggest to replace IPR with RES (airborne radio-echo sounder). It is the term most often used in literature by now and also extended its meaning from the initial analog systems to modern multi-antenna phase-sensitive systems.
* * *
Author: oeisen     Subject: Hervorheben     Date: 11.03.23, 13:32:05

commensurately

This is not true for ultrawide band radars such as MCORDS5 - they can resolve firn layers of 1 m resolution.
* * *
Author: oeisen     Subject: Hervorheben     Date: 11.03.23, 13:32:40

Earth

**Page: 3**

GOG3
explain/write out once

/bandwidth
a bit unclear like this - rather
dual-frequency (i.e. different bandwidth)

Author: oeisen    Subject: Hervorheben    Date: 11.03.23, 13:37:20

Mention elevation contours in caption (i.e. from 600 m to ... m every 200 m).
I suggest to change dashed black to solid black, as dashed is particular unclear for zone IIb.

Author: oeisen    Subject: Hervorheben    Date: 11.03.23, 13:38:21

with

Author: oeisen    Subject: Hervorheben    Date: 11.03.23, 13:38:44

shouldn't this be CReSIS?
* * *
Author: oeisen       Subject: Hervorheben       Date: 11.03.23, 13:40:09

unclear. Suggested rewrite:
the radar return from the surface is influenced to a depth
* * *
Author: oeisen       Subject: Hervorheben       Date: 11.03.23, 13:43:24

relative or not?
* * *
Author: oeisen       Subject: Hervorheben       Date: 11.03.23, 13:48:23

below, k is the wavenumber. Chose a different letter here (please do not use k_w)
* * *
Author: oeisen       Subject: Hervorheben       Date: 12.03.23, 10:00:08

effective
* * *
Author: oeisen       Subject: Sticky Note  Date: 12.03.23, 10:01:41

later you use the ordinary relative permittivity. Please specify in the text, which one eps_eff denotes. Distinguish between absolute an relative permittivity by using subscript _r, if necessary.
* * *
Author: oeisen       Subject: Hervorheben       Date: 11.03.23, 13:44:57

ordinary relative permittivity
* * *
Author: oeisen       Subject: Hervorheben       Date: 11.03.23, 13:45:39

why between?
Rather of?

Author: oeisen     Subject: Hervorheben     Date: 11.03.23, 13:46:38

permittivity changes
* * *
Author: oeisen     Subject: Hervorheben     Date: 12.03.23, 10:32:11

surface reflection coefficient r:
power or amplitude? I assume amplitude, but clarify.
* * *
Author: oeisen     Subject: Hervorheben     Date: 11.03.23, 13:50:12

rms height: calculated over particular window length, all profiles or other?
Please specify.
* * *
Author: oeisen     Subject: Hervorheben     Date: 11.03.23, 13:47:38

k is the wavenumber

You must not use the same variable for two different purposes in the same manuscript - here windowing factor and wavenumber.
* * *
Author: oeisen     Subject: Hervorheben     Date: 11.03.23, 13:49:12

laser: specify: laser altimetry or laser scanning?
I assume airborne laser, please clarify
* * *
Author: oeisen     Subject: Hervorheben     Date: 11.03.23, 13:54:00

0.22 dB to Pc

Could you give a percentage of the average value of Pc? At this stage the reader did not see any Pc value, so does not know how (in)sigificant this is and later you say it is "conservative". Sufficient to say e.g.
"0.22 dB to Pc, i.e. less than x% in terms of dB."
* * *
Author: oeisen     Subject: Hervorheben     Date: 11.03.23, 13:50:41

sheet - this is an ice cap - replace
* * *
Author: oeisen     Subject: Hervorheben     Date: 11.03.23, 13:51:07

freshly fallen snow
* * *
Author: oeisen     Subject: Hervorheben     Date: 11.03.23, 13:51:39

can you further specify why? E.g. any data which indicate that? Please clarify

Author: oeisen    Subject: Hervorheben    Date: 12.03.23, 09:54:33
unlcear: should be along HiCARS2 transects. Please clarify.

Author: oeisen    Subject: Hervorheben    Date: 12.03.23, 09:55:42
picking the firn-ice interface in

Author: oeisen    Subject: Hervorheben    Date: 12.03.23, 09:59:35
ordinary relative permittivity

Author: oeisen        Subject: Hervorheben        Date: 12.03.23, 10:03:38

simply

"firn" layers

Author: oeisen        Subject: Hervorheben        Date: 12.03.23, 10:06:01

Please explain operators:
"The operator ||...|| denotes ... IFFT is the ..."

Author: oeisen        Subject: Hervorheben        Date: 12.03.23, 10:06:33

no tapering used?

Author: oeisen      Subject: Hervorheben      Date: 12.03.23, 10:09:05

as mentioned previoulsy consider to change dashed to solid black.

Add:
"Background and contours as in Fig. 1".
in fact you could also write
"Background, contours and firn boundaries as in Fig. 1." and remove the separate description of the firn boundary here.

Author: oeisen          Subject: Hervorheben          Date: 12.03.23, 10:22:41
aren't units needed for IQR in legend?

During copy-editing it might unfortunately be suggested again to put all three panels on top of each other to fill only one column in the final typeset version - in contrast to the reviewer's suggestions. I find the comparability in the previous figure version also more compelling than now.

Author: oeisen    Subject: Hervorheben    Date: 12.03.23, 10:15:41

of what?
the spatial distribution of the ratio?

Author: oeisen    Subject: Hervorheben    Date: 12.03.23, 10:17:31

This is ambiguous.
Pc is in dB
Pn is in dB
so their ratio would at first sight be unitless.

Or do you rescale the ratio again to dB by taking the logarithm? Please clarify, as also important for the figures indicating Pc/Pn

Author: oeisen    Subject: Hervorheben    Date: 12.03.23, 10:19:15

firn at the surface

I consider it important to clarify that satellite imagery/measurements can only indicate the properties at the surface, but not below (eg if there is left-over firn below an ice slab).

Author: oeisen    Subject: Hervorheben    Date: 12.03.23, 10:25:49

the surface signal probes

Author: oeisen        Subject: Hervorheben        Date: 12.03.23, 10:30:38
section Discussion).

Author: oeisen      Subject: Hervorheben      Date: 12.03.23, 10:36:31

resolutions of the surface reflection (i.e., z0)

Author: oeisen          Subject: Hervorheben          Date: 12.03.23, 10:42:03
Figure: as for previous figures, please add info on elevation contours in caption.

Regarding my comment in previous figures for the dashed black lines, the dash spacing here is small enough to indicate clearly the boundaries, whereas it is too wide in the previous figures.

Author: oeisen          Subject: Hervorheben          Date: 12.03.23, 10:44:44

Quite a long subscript. I suggest to put the radar system as a superscript instead to increas readibility.

also the

Author: oeisen    Subject: Hervorheben    Date: 12.03.23, 10:52:35

editor,

Author: oeisen    Subject: Hervorheben    Date: 12.03.23, 10:52:55

xxx.

---

## Author Response (AR2)

**Spatial characterization of near-surface structure and meltwater runoff conditions across Devon Ice Cap from dual-frequency radar reflectivity**
Chan et al.

**Response to comments from Editor:**

General comment:

*Ice-penetrating radar: I understand your wish to keep this terminology, but keep in mind that IPR is nor very often used. For comparison, IPR about 2 million times compared to radio-echo sounding 30 million times in a common search engine. I support the statement of the reviewer here. In fact, we recently argued to get rid of this term altogether (Schlegel et al., Ann. Glac., 2023).*
*Ice-penetrating is nothing else than ground-penetrating, where the ground is made of ice (but could also be dry sand or alike). However, GPR is usually referred to as ground-based. This distinction now becomes obsolete as GPRs are also flown underneath helicopters.*
*To be more consistent with the most used convention I suggest to replace IPR with RES (airborne radio-echo sounder) or simply radar. RES is the term most often used in literature by now and also extended its meaning from the initial analog systems to modern multi-antenna phase-sensitive systems, for instance, basically inherent in the name of HiCARS, MCoRDS or MARFA. None of the systems you use in your study carries the denomination "ice" in its name. So why do you want to keep IPR?*

> Thank you for the comment. We understand the desire to adopt a common terminology for these radar systems. Given that the Schlegel et al. paper is not yet published and publicly available, we cannot evaluate the arguments that favor one term over another for these systems. Moreover, we believe that ice-penetrating radar, compared to radio echo sounding, is more appropriate for this work, because we only utilize the surface return to investigate near-surface properties and are not sounding to subsurface depths. The comments from the reviewers were focused on distinguishing between airborne and ground-based radar, which we initially referred to as IPR vs. GPR. That has since been changed to clarify when we are referring to airborne ice-penetrating radar vs. surface-based radar. While the airborne radars HiCARS, MARFA, and MCoRDS don't carry the term "ice" in them, they were designed with the intent to study ice, and we would like to reflect their main purpose when referring to them in this study as ice-penetrating radar. We also added "also known as radio echo sounding" when first introducing "ice-penetrating radar" in the manuscript.

*The second general comment addresses the usage of operators & variables, which is partly ambiguous. I would ask you to check thoroughly and please clarify where marked in the annotated manuscript (e.g. epsilon (absolute or relative), k, ||...||, ...).*

> We reviewed the operators and variables in the manuscript and clarified their usage.

*Colour scale: (e.g. Fig. 2) Rainbow is the worst color scale in general one could use, not only for color-blind people, but also to indicate changes. Nothing to change here now, but keep in mind for future publications. See e.g.*
*https://eos.org/features/visualizing-science-how-color-determines-what-we-see*

> Thank you for the note. The color scale used in this figure is actually Turbo, which is an improvement over the traditional Rainbow. We found it to be a good balance between a linear color scale capable of capturing changes in Pc and being colorblind-friendly in most cases (https://ai.googleblog.com/2019/08/turbo-improved-rainbow-colormap-for.html).

*Figures: Boundary of firn zones: I find the dashed black line makes it partly difficult to see the line correctly. Consider changing to solid black or decrease the gap between dashed (e.g. fine in Fig. 5) Explain this also in the new captions that gray lines indicate elevation contours, only shown for 600 m and higher. Dashed black lines indicate firn region boundaries.*

We clarified that thin gray lines indicate elevation contours in the captions. We also revised the firn zone boundaries by decreasing the spacing between dashes, similar to Fig. 5.

Specific comments (P = page, Number = line in annotated author track changes pdf):

P1

*12: see general comment - change:*
*airborne radar or airborne radio-echo sounding*
       Please see above.

*23: higher*
*I don't get this argument in the abstract, because not mentioned before.*
*Undefined, what higher frequency means here, can't be HF. Do you mean UHF?*
*Or rather high-resolution (cm-dm scale)?*
       We changed this to high resolution.

P2

*68/69: This is not true in general, e.g. for ultrawide band radars such as MCORDS5 - they can resolve*
*firn layers of 1 m resolution. I would rephrase to be less strong.*
       We rephrased this sentence to be less strong and more focused on previous data collected.

*75: earth -> Earth*
       We made this edit.

P3

*90: GOG3: explain/write out once*
       We defined this acronym in the revised manuscript.

*96: /bandwidth*
*a bit unclear like this - rather dual-frequency (i.e. different bandwidth)*
       We reworded this to clarify that we are using airborne radars at two different frequencies and bandwidths for this work.

P4

*120: Mention elevation contours in caption (i.e. from 600 m to ... m every 200 m). I suggest to change*
*dashed black to solid black, as dashed is particular unclear for zone IIb.*
       We mentioned the elevations contours in the caption and decreased the spacing between dashes similar to Figure 5.

*130: by -> with*
*130: shouldn't this be CReSIS?*
       We changed 'by' to 'with' and added CReSIS.

P5

*145: unclear. Suggested rewrite: "the radar return from the surface is influenced to a depth"*
       We made this suggested edit.

*150: below, k is the wavenumber. Chose a different letter here (please do not use k_w)*
We simplified this equation to express it in terms of wavelength instead of wavenumber.

*150: relative or absolute effective permittivity?*
*later you use the ordinary relative permittivity. Please specify in the text, which one eps_eff denotes.*
*Distinguish*
*between absolute a relative permittivity by using subscript _r, if necessary.*
*163: ordinary relative permittivity*
All permittivities mentioned in the manuscript are relative. We clarified this in the text, where
appropriate.

170: why between? Rather "of"?
We changed this to "of".

P6

*201: change to "permittivity changes"*
We made this change.

*209: surface reflection coefficient r: power or amplitude? I assume amplitude, but clarify.*
We clarified that this is in amplitude.

*209: rms height: calculated over particular window length, all profiles or other? Please specify. (I know*
*it was there but deleted, but a bit more info is required.)*
We specified that this is calculated over the wavelength scale.

*209: now k is the wavenumber. You must not use the same variable for two different purposes in the same*
*manuscript - here windowing factor and wavenumber. Please clarify.*
We kept k for windowing factor and removed wavenumber.

*211: laser: specify: laser altimetry or laser scanning? I assume airborne laser, please specify.*
We clarified that it's from laser altimetry.

*213: "0.22 dB to Pc": Could you give a percentage of the average value of Pc? At this stage the reader*
*did not see any Pc value, so does not know how (in)sigificant this is and later you say it is "conservative".*
*Sufficient to say e.g. "0.22 dB to Pc, i.e. less than approximately x% in terms of dB."*
It's about 1% of the average Pc value, and we added this in the manuscript.

*214: sheet - this is an ice cap – replace*
We replaced this with ice cap.

*214: change to "freshly fallen snow".*
We changed this to freshly fallen snow.

*214: assume: can you further specify why? E.g. any data which indicate that? Please clarify*
Both the GOG3 and SRH1 surveys were conducted in the spring before the melt season. Thus,
surface roughness would be mainly influenced by the presence of snow, which we assume to be
the case for both surveys. We clarified this in the text.

P7

*262: unlcear: should be along HiCARS2 transects. Please clarify.*
  We change "collocated with" to "along".

*266: add: "picking the firn-ice interface in"*
  We added "picking the firn-ice interface".

*288: gain specify: "ordinary relative permittivity"*
  We specified that this refers to relative permittivity.

P8

*321: I suggeset to simply write "firn" layers*
  We simplified this to "firn" layers.

*327ff: Please explain operators: "The operator ||...|| denotes ... IFFT is the ..."*
  We explained the operators of Eq. 4.

*332: no tapering used?*
  This is an analytical model where the reflection coefficient is calculated at every frequency contained in the chirp, and the maximum is chosen after being convolved with the chirp. Thus, tapering is not applicable here, although the effects of tapering are captured in the actual airborne radar data with the windowing factor, resulting in a degradation of the vertical resolution.

P9

*342: consider to change dashed to solid black.*
  We decreased the spacing between dashes.

*342: Add: "Background and contours as in Fig. 1". In fact you could also write "Background, contours and firn boundaries as in Fig. 1." and remove the separate description of the firn boundary here.*
  We referred to Fig. 1 and removed the separate description of the boundaries as suggested.

P10

*500: aren't units needed for IQR in legend?*
  We added the units for IQR in the legend.

*500_ During copy-editing it might unfortunately be suggested again to put all three panels on top of each other to fill only one column in the final typeset version to save space - in contrast to the reviewer's suggestions. I find the comparability in the previous figure version also more compelling than now.*
  We included the figure as is but can rearrange the plots as needed during copy-editing.

P11

*529: inspection of what? The spatial distribution of the ratio?*
  Yes, we specified this in the text.

*529: Pc/Pn: This is ambiguous.*

*Pc is in dB*
*Pn is in dB*
*so their ratio would at first sight be unitless.*
*Or do you rescale the ratio again to dB by taking the logarithm? Please clarify, as also important for the*
*figures indicating Pc/Pn*

  We specified that the Pc/Pn ratio is expressed in dB and calculated by taking the difference
  between Pc and Pn in logarithmic space.

*534: firn at the surface*
*I consider it important to clarify that satellite imagery/measurements can only indicate the properties at*
*the surface, but not below (eg if there is left-over firn below an ice slab).*

  We clarified that the images indicate the lack of firn at the surface.

*554: the surface signal probes*

  We clarified that it's the surface signal.

P13

*648: add "section Discussion)."*

  We changed it to Sec. 4.1.

P14

*661: resolutions of the surface reflection (i.e., z0)*

  We specified the surface return in the caption.

P16

*698: Figure: as for previous figures, please add info on elevation contours in caption. Regarding my*
*comment in previous figures for the dashed black lines, the dash spacing here is small enough to indicate*
*clearly the boundaries, whereas it is too wide in the previous figures.*

  We added the elevation contours info in the caption.

P17

*709: Quite a long subscript. I suggest to put the radar system as a superscript instead to increase*
*readibility.*

  We changed all the subscripts, here and in the text, to place the radar system in the superscript.

P19

*796: add "also" before "the first proof"*

  We added "also".

*874: Please don't forget to provide final publication number during copy-editing.*

  We added the final publication number.

SUPPLEMENT:

*P2 Title x-axis Fig S1: Density should be (kg m^-3)*
> We changed the units to kg m^-3.

*P3 Figure S2*
*Either use a) and b) or add explicitly which panel you are referring to, e.g. solid gray lines (a/top).*
*Explain why the ration Pc/Pn is still in dB (the comment in main).*
*See main: zone boundaries: they are hard to follow and look a bit chaotic. Please improve visualisation, e.g. by having solid black lines?*
*Add: "thin grey lines are elevation contours (m).*
*IQR: how about units for values in legend?*
> We used solid black lines for the old zone boundaries and decreased the spacing between the dashes for the new zone boundaries. We added info about the elevation contours and units for IQR in the legend. Explanation for the Pc/Pn ratio was added to the main text. We reorganized the caption, such that the first part refers to the top subplot, and the remaining part refers to the bottom subplot.

*P4 Figure S3: add: HiCARS2/MARFA (top) ... MCoRDS3 (bottom)*
> We added (top) and (bottom).

*P5 Figure S4: • .• - delete blank in 1. 8.*
> We deleted the blank in between "." and "8".